



# Glaciological and meteorological monitoring at LTER sites Mullwitzkees and Venedigerkees, Austria, 2006-2022

Lea Hartl[1,2], Bernd Seiser[1], Martin Stocker-Waldhuber[1], Anna Baldo[1,3], Marcela Violeta Lauria[1,3], and Andrea Fischer[1]

[1]Institute for Interdisciplinary Mountain Research, Austrian Academy of Sciences, Innrain 25, 6020 Innsbruck, Austria
[2]Alaska Climate Research Center, University of Alaska Fairbanks, 2156 Koyukuk Drive, Fairbanks, AK 99775, USA
[3]Department of Atmospheric and Cryospheric Sciences, University of Innsbruck, Innrain 52, 6020 Innsbruck, Austria

**Correspondence:** Lea Hartl (lea.hartl@oeaw.ac.at)

**Abstract.** Glaciers in the Alps are losing mass at unprecedented and accelerating rates. Monitoring of glacier mass change as well as relevant atmospheric parameters plays an important role in improving understanding of local and downstream impacts. We present subseasonal, seasonal, and annual glaciological mass balance data and meteorological observations from Mullwitzkees and Venedigerkees, two glacier monitoring sites in the Hohe Tauern range of the Austrian Alps. Ablation stake

networks were established on Mullwitzkees in 2006/07 and on Venedigerkees in 2011/12. Monitoring is ongoing. In addition to stake readings at subseasonal intervals, accumulation measurements (snow pits and probing) are carried out seasonally. The glaciological data set consists of subseasonal floating date measurements as well as fixed date seasonal and annual values. Fixed date glacier wide mass balance was derived from annual point mass balance values. Automatic weather stations measuring standard meteorological parameters were installed near Mullwitzkees and Venedigerkees in 2020 and 2019, respectively.

Meteorological data is provided in 10 minute intervals. Uncertainties for individual point mass balance measurements were computed following the approach of the Swiss glacier monitoring service (GLAMOS), taking into account estimated density and reading errors. The subseasonal mass balance records highlight shorter term variability in mass loss and the linkage with meteorological conditions. The most negative annual point mass balance recorded in the period of record was -5.8 ± 0.66 m w.e. at an elevation of 2536 on Venedigerkees. 2022 stands out as the most negative mass balance year to date in both

time series, particularly at higher elevations. The cumulative specific mass balance (glacier wide) over the period of record was -14.68 m w.e. at Mullwitzkees and -8.79 m w.e. at Venedigerkees. Data is available in PANGAEA publication series and the associated datasets. The main publication series are updated annually. The Mullwitzkees mass balance datasets can be found at: doi:10.1594/PANGAEA.965660 and doi:10.1594/PANGAEA.965719. The Venedigerkees data can be found at doi:10.1594/PANGAEA.965648 and doi:10.1594/PANGAEA.965729.

# 1 Introduction

Environmental monitoring in data sparse high mountain areas is essential to assess changing natural and human systems under ongoing and accelerating climate change (Collins et al., 2011; Costa et al., 2016; Bojinski et al., 2014; Gaiser et al., 2020; Zemp et al., 2021). The mountain cryosphere is an integral part of the overall mountain environment and strongly





linked to atmospheric processes, which drive cryospheric changes in the short term as well as on climatological time scales
(Hock and Huss, 2021). Glacier mass change in turn is an important climate indicator and drives hydrological change across
scales, from locally shifting runoff regimes in single catchments to global sea level rise (Huss and Hock, 2018; Zemp et al.,
2019). As an Essential Climate Variable (ECV, Bojinski et al. (2014)), glacier mass balance is commonly incorporated into
environmental monitoring programs in mountain areas alongside related meteorological parameters (Collins, 2022). The glacier
wide and elevation zone mass balance data of Mullwitzkees (MWK) and Venedigerkees (VK) have partially been described in
overview publications related to glacier monitoring in Austria (Stocker-Waldhuber et al., 2013; Fischer et al., 2014; Hansche
et al., 2023) and in non-peer reviewed reports submitted annually to the respective funding agencies (accessible through the
PANGAEA publication series Stocker-Waldhuber et al. (2016) and Seiser and Fischer (2016)). Intermediate and point data
have not previously been published or described.

Measuring glacier surface mass balance via the glaciological method, i.e. with ablation stakes and snow pits, coring, or
probing to quantify snow accumulation, has a long tradition due to the relative simplicity of the method (Cogley et al., 2011;
Zemp et al., 2015; Fischer et al., 2018). Standardized mass balance observations are collected and compiled by the World
Glacier Monitoring Service (Zemp et al., 2023), often in the form of seasonal or annual glacier wide mass balance, which is
spatially inter- or extrapolated from point measurements (Østrem and Brugman, 1991; Cogley et al., 2011). Various systems
for defining the respective measurement periods are in use and the duration of a mass balance season or year may adhere to
fixed dates (start and end of the hydrological year), floating dates, or stratigraphic notations (Cogley et al., 2011). Depending on
the measurement system, temporal extrapolation is required in addition to spatial extrapolation to arrive at seasonal or annual
glacier wide mass balance from point measurements acquired at discrete moments in time and space.

Point mass balance data form the basis of glacier wide mass balance estimates (e.g., Østrem and Brugman (1991); Cogley
et al. (2011); Geibel et al. (2022)). Preserving such data with appropriate metadata and at the greatest possible temporal resolu-
tion is essential for potential future reanalysis and homogenization of time series (Zemp et al., 2013). Additionally, point mass
balance data can support calibration and validation of mass balance models (Schaefli and Huss, 2011). Subseasonal observa-
tions improve understanding of short term mass changes, e.g. due to extreme events like heat waves (Cremona et al., 2023), and
contribute to the improved quantification of relationships between atmospheric parameters and mass change (Vincent et al.,
2017), particularly when combined with meteorological observations.

In Austria, 13 glaciers (including MWK and VK) were subject of systematic mass balance monitoring as of the 2021/22
season (Hansche et al., 2023). Measurements at eight additional glaciers have been conducted in the past but were discontinued.
Hansche et al. (2023) provide an overview of current glacier monitoring programs in Austria, listing coordinating organizations
and funding agencies for each site. Spatially aggregated glacier wide mass balance from all current Austrian monitoring sites
can be accessed through the WGMS database (WGMS, 2023). Point mass balance has typically not been published along with
the glacier wide data.

In this publication, we present glaciological and meteorological data from Mullwitzkees (MWK) and Venedigerkees (VK),
two glaciers in the Hohe Tauern range of the Austrian Alps. Both sites are part of the Long Term Ecological Research program
(LTER, Collins (2022)). We focus on a comprehensive compilation of the subseasonal point mass balance data available from



MWK and VK. In the interest of comparability with other datasets, we follow the approach of the Swiss Glacier Monitoring Program (GLAMOS) regarding data structure, metadata parameters, and uncertainty assessments (GLAMOS, 2021; Geibel et al., 2022). In addition to the subseasonal mass balance data, we include winter and annual fixed date mass balance at the respective measurement points, as well as extrapolated to elevation zones and the glacier scale. For the most recent 3-4 years of the periods of record, the glaciological data is complemented by records of temperature, humidity, atmospheric pressure, wind, radiation and precipitation parameters from two automatic weather stations (AWS) located in approximately 500 m and 800 m distance to MWK and VK, respectively.

## 2 Data and methods

### 2.1 Study Sites

MWK and VK are temperate, winter accumulation type glaciers in the core zone of Hohe Tauern National Park (HTNP). HTNP is Austria's largest national park and is shared between the states of Salzburg, Tyrol, and Carinthia. Abbout a third of all Austrian glaciers (337 glaciers, Fischer et al. (2015a)) are located within the boundaries of HTNP. MWK and VK are situated in the western section of HTNP in the Venediger Range, the mountain group around Großvenediger (3657m, Fig. 1). Großvenediger and VK are also within a Natura 2000 protected area as per the European Union Habitats Directive (European Commission and Directorate-General for Environment and Sundseth, 2012; European Commission and Directorate-General for Environment and Sundseth, 2015).

MWK is on the Tyrolean side of Großvenediger about 3 km southeast of the summit. The upper part is also known as Äußeres Mullwitzkees while the lower sector is known as Zettalunitzkees. We use the name Mullwitzkees (MWK) to include both parts. MWK can be classified as a plateau glacier and as a valley glacier. In 2018, it covered an area of 2.56 km$^2$ and an elevation range from 2730 m to 3400 m. MWK is predominantly south-facing and located on the southern side of the main Alpine Divide. The watershed drains locally into the Dorferbach and subsequently into the rivers Isel and Drava and eventually the Danube. Ice thickness surveys carried out in the context of the second Austrian Glacier Inventory (GI) indicated that the glacier is relatively thin, with ice thickness values of about 50-70 m at the time of measurement in 2003 (Fischer and Kuhn, 2013; Fischer et al., 2015). The thickest ice is found in the central part of the plateau. Ice thickness decreases towards the highest elevations of the glacier. Rock outcrops have recently appeared in this area, confirming the limited ice thickness. MWK borders on Rainerkees in the west, Schlatenkees in the north, and Frosnitzkees in the east. The glacier boundaries are defined by the respective ice divides.

Mass balance measurements at MWK began in 2006 as part of wider hydrological and glaciological monitoring efforts by HTNP and the department for hydrology of the state of Tyrol (HD Tyrol, *Hydrographischer Dienst der Abteilung Wasserwirtschaft, Amt der Tiroler Landesregierung*). HTNP and HD fund the monitoring program. Runoff is measured by HD Tyrol in the Dorferbach and available through the data platform ehyd (212233, Hinterbichl (Mühle)/Dorferbach, https://ehyd.gv.at/). An AWS was installed near MWK in August 2020.



VK is a valley glacier extending from 2503 m to 3400 m over an area of 1.82 km$^2$ in 2018. The main section of the accumulation zone is directly below the summit ridge of Großvenediger, in the state of Salzburg. The upper part of the glacier is north facing. The tongue turns to the west and southwest below about 2900 m. VK is a former tributary of Obersulzbachkees, which retreated strongly in recent years and has disintegrated into multiple individual glaciers (2009, GI3; Fischer et al. (2015a, b)). After VK became disconnected from Obersulzbachkees, the name Venedigerkees (VK) was established for the easternmost remaining glacier, while the part directly north of the summit has since been called Sulzbacherkees (SK). Initially, three ablation stakes were maintained on the tongue of SK in addition to the stake network on VK but this has been abandoned due to the rapid retreat of SK. VK has a long ice divide with Untersulzbachkees to the east and drains into the Obersulzbach valley and subsequently into the rivers Salzach, Inn, and Danube.

The mass balance monitoring program at VK began in 2012 and is funded by the department for hydrology of the state of Salzburg (HD Salzburg, *Hydrographischer Dienst des Landes Salzburg*). Runoff is measured by HD Salzburg in the Obersulzbach and available through the data platform ehyd (203893, Kees/Obersulzbach, https://ehyd.gv.at/). Since August 2019, an AWS near VK is used to monitor meteorological parameters in addition to mass balance.

### 2.2 Glaciological parameters

#### 2.2.1 Point mass balance

To measure accumulation, snow depth and stratigraphic bulk density (measurements of approx. 20 cm layers averaged over the depth of the snow pit) from the snow surface to the firn or ice horizon were measured at three to four snow pits at each glacier in spring. Snow depth at other locations was measured by probing. The water equivalent value at the probing locations was computed based on the probed depth and the density measurements at the snow pits. The accumulation surveys were carried out as close as possible to April 30, the midpoint of the hydrological year and the fixed date for winter mass balance.

To measure ice ablation, networks of ablation stakes were installed at MWK in 2006 and VK in 2012. The number of stake locations per glacier ranged from 10 to 17 at MWK and 13 to 16 at VK during the respective time series. For initial installation as well as ongoing monitoring, wooden ablation stakes were drilled 6-12 m into the ice with a steam drill. Height change of the stakes was measured at irregular intervals throughout the ablation season. Ice ablation at the stakes was obtained by multiplying the height change with an assumed ice density of 900 kg/m$^3$ (Cogley et al., 2011).

In fall, the final stake readings of each season took place as close as possible to September 30, the end of the hydrological year. Snow pits were dug and density was measured at the same pit locations as in spring if snow remained at these locations. Probing was similarly carried out if snow was present.

Each stake reading and snow pit measurement provides an intermediate mass balance observation for the period between the measurement date and the date of the last known value for the location. We derived a fixed date annual and seasonal point mass balance at each measurement location (Kaser et al., 2003; Cogley et al., 2011), i.e. the closest observations to the end of the hydrological year (September 30) and the end of the accumulation season (April 30) are extrapolated to the respective fixed dates. The temporal extrapolation is informed by meteorological data such as snow fall between the measurement and the

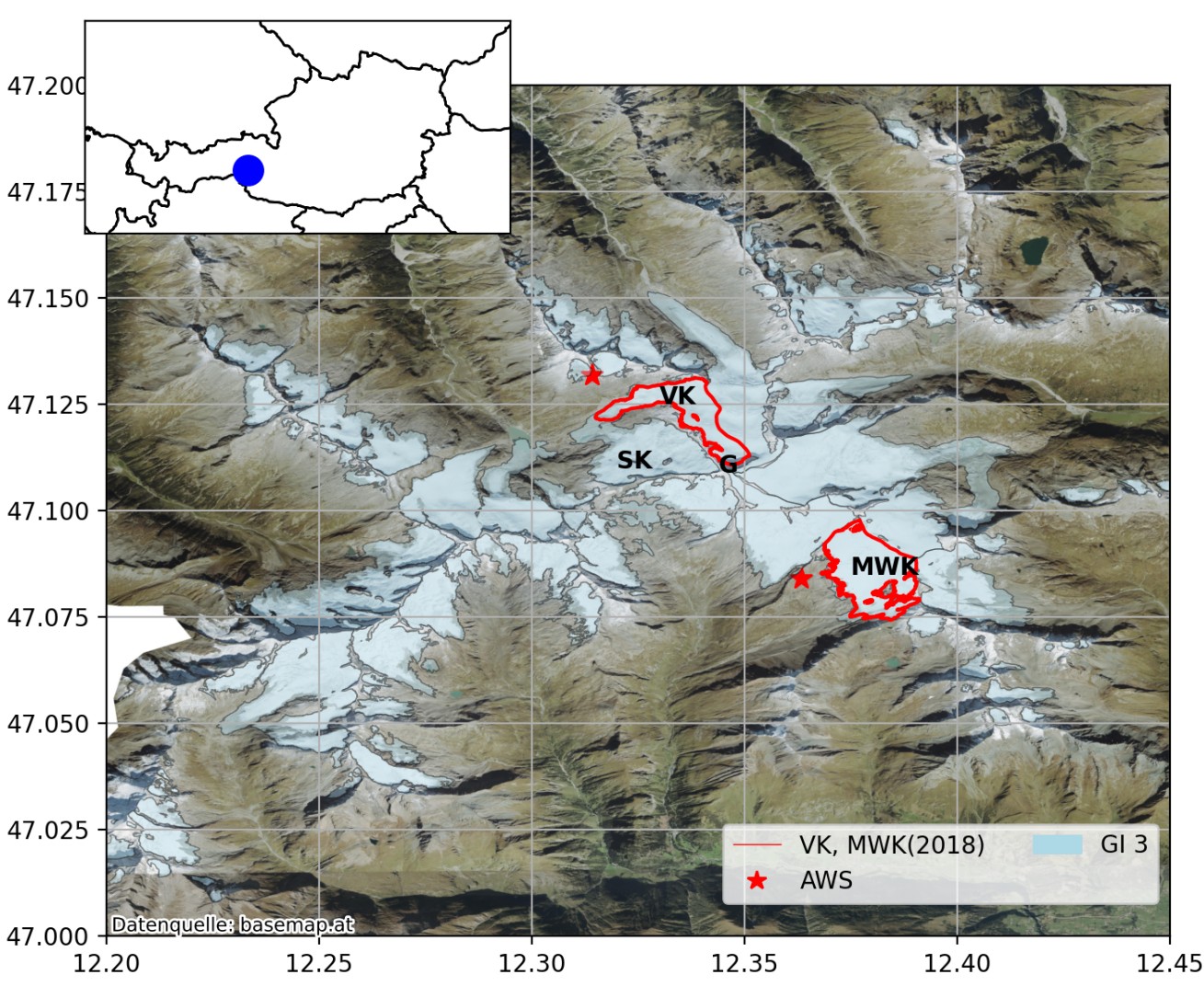

**Figure 1.** Glaciers in the Venediger range (outlines: third Austrian glacier inventory, 2009, Fischer et al. (2015a)). Outlines of VK and MWK for 2018 in red. Stars mark the two automatic weather stations. SK: Sulzbacherkees. G: Großvenediger, summit. Background: basemap.at. The inset shows the location of the study sites in Austria.



fixed date. In case of snow accumulation between the date of the last measurements and the fixed date, snow water equivalent
(SWE) is estimated and accounted for in the annual fixed date value. Similarly, SWE gained or lost between the spring snow
survey and the spring season fixed date is estimated to derive the fixed date winter point mass balance. The seasonal and annual
fixed date point mass balance values are given in water equivalent units (mm w.e.). The first ablation measurement period of a
given hydrological year begins at the start of that hydrological year (October 1) and refers to the fixed date ablation reading of
that date.

Locations of stakes, snow pits, and snow depth probing were measured with a handheld GPS device. For the stakes, GPS
coordinates were not recorded during every reading and the latest GPS measurement in each hydrological year is considered
that year's location for the respective stake. The elevation of measurement locations was not recorded in the field and was
extracted from digital elevation models (DEMs) based on the coordinates of the stake. The respective DEMs were acquired
in 2008 for MWK and 2012 for VK. Throughout the period of record, the position of individual stakes changed due to ice
movement as well as occasionally due to repositioning during maintenance activities. Repositioning of stakes was avoided as
much as possible. Reasons for repositioning were mostly related to newly formed crevasses or area loss, i.e. stakes were moved
away from crevasses or the ice edge to maintain safe access and representative measurement locations.

In the following, observations of height change of the ice surface and corresponding ablation values between consecutive
readings as well as observations of snow height are referred to as **intermediate measurements** (Geibel et al., 2022). **Annual
and seasonal** (winter season) measurements refer to values derived from cumulative intermediate measurements and extrapo-
lated temporally to fixed dates. The intermediate data files contain the intermediate measurements acquired on variable dates.
The annual and seasonal data files contain point mass balance values extrapolated to fixed dates from the closest available
measurement dates and can be considered a derivative product of the intermediate files. Table 1 shows the total and average
number of different measurement types available for each site.

In the interest of comparability and compatibility with mass balance data from other sites, the point mass balance data and
metadata were formatted to be consistent with the structure introduced in detail by Geibel et al. (2022) and applied in the Swiss
GLAMOS program (GLAMOS, 2021). We adhere to their quality flags as closely as possible for our intermediate dataset and
use the respective assigned errors as given in Geibel et al. (2022) and the documentation of the associated data publication
(GLAMOS, 2021) unless otherwise stated. For the intermediate data, the following quality flags were applied:

– **Date quality:** the start and end dates of each period are known for ablation measurements at the stake locations (flag 0).
In some cases, measurements were carried out on two consecutive days and the exact date on which specific stakes were
surveyed was not recorded. In these instances, the second date is stated and the date is still considered known. For snow
surveys in spring (probing and snow pits), the true start date of the measurement period (i.e. the start of seasonal snow
accumulation) is unknown but estimated as the start of the hydrological year in adherence with the fixed date system.
Therefore, October 1 of the previous year is entered as the start date of the period and the date quality flag is set to
"estimated/unknown" (flag 3). Snow surveys in fall may include snow that fell during the previous accumulation season
and/or new snow that fell recently. The start date is considered "unknown" and flagged accordingly without an estimated
start date. Following GLAMOS (2021), the error associated with estimated or unknown dates is assumed to be elevation





dependent with 36 cm between 1500 and 2500 m, 31 cm between 2500 and 3000 m, and 22 cm above 3000 m. This error contributes to the reading uncertainty.

- **Position quality:** For most stake measurements in our data set, the position was estimated from previous measurements and flagged as such (flag 4). When the date of the GPS data point is known, as is usually the case for snow pit and probe data, the "position quality" flag is adjusted accordingly. This error does not contribute to the reading uncertainty (Geibel et al., 2022).

- **Density quality:** Ice density is assumed to be 900 kg/m$^3$ (flag 1, error: 2% of mass balance). Snow density is measured at snow pit locations (flag 2, error: 5% of mass balance). This error contributes to the mass balance uncertainty.

- **Measurement quality:** set to "typical uncertainty for the specific method" (flag 1, error: 0).

- **Measurement type:** The intermediate data set contains ablation stake readings (flag 1, 5 cm type error) and snow pit data (flag 2, 10 cm type error). The measurement type is always known. This error contributes to the reading uncertainty.

- **Special cases:** The winter balance for 2011/12 at VK was reconstructed in spring 2012/13 by digging snow pits to the previous year's firn horizon. The respective snow pit values are flagged with measurement quality flag 5 for "reconstructed value".

For fixed date values (annual and winter balance), the quality flags and associated uncertainties were adjusted as follows to account for fixed vs. floating dates:

- **Date quality:** Dates are considered known (start, mid point, and end of the hydrological year).

- **Position quality:** The position of measurement points was estimated from previous measurements and flagged as such (flag 4).

- **Density quality:** The annual fixed date density and derived SWE values at the stakes are based on a combination of ice density and fresh snow at the stake locations and flagged accordingly (flag 5). The associated error is set to 10% of mass balance in mm w.e. For winter and autumn snow pit data, the density flag is set to "Density of snow estimated from nearby measurement" (flag 3, 8% of mass balance). For snow probing measurements, the density was estimated from the snow pit data and also set to "Density of snow estimated from nearby measurement".

- **Measurement quality:** For all fixed date values, the measurement quality flag is set to 5, with an associated quality error of 30 cm that contributes to reading uncertainty. This is the equivalent of the quality flag "reconstructed value (other reason)" in the system employed by GLAMOS (2021). The intention is to indicate that fixed date values were temporally inter- or extrapolated ("reconstructed") from the intermediate values.

- **Measurement type, probing:** In addition to stakes and snow pits (as in the intermediate data set), the fixed date data set includes a separate file with snow depth probing data. The measurement type is set to 3 for these data points, with an



associated 15 cm type error. Note that this data type and flag differs from the GLAMOS system. The probe data is only available as water equivalent values extrapolated from the date of measurement to the fixed date. Records of the raw data (snow depth on the day of measurement) were not kept. Probe data has an additional flag that indicates whether or not the water equivalent value of a given probe data point is within ±100 mm w.e. of the spatially integrated mass balance value at the probing location. This aims to identify probe data that is not representative for winter or annual point mass balance. For the annual data, values flagged as not representative are typically due to new snow that fell shortly prior to the measurement in the ablation zone. During spring surveys it is often challenging to definitively identify the previous year's firn horizon by probing. In addition to spatial inhomogeneity of snow depth, probe data may have systematic biases in a given year due to, e.g., ice lenses in the snow pack that cannot be penetrated with the probe.

Following Geibel et al. (2022), the uncertainty of each reading ($\sigma_r$) is then assumed to be:

$$\sigma_r = \sqrt{\sigma_{date}^2 + \sigma_{mtype}^2 + \sigma_{mqual}^2} \text{ [eq 1]}$$

where $\sigma_{date}$ is the date uncertainty, $\sigma_{type}$ is the measurement type uncertainty, and $\sigma_{mqual}$ is the measurement quality uncertainty. The mass balance (b) uncertainty is:

$$\sigma_{mb} = \sqrt{\sigma_r^2 + \sigma_{density}^2} * b \text{ [eq 2]}$$

$\sigma_{density}$ is given as a percentage of the mass balance and hence multiplied with b. We refer to Geibel et al. (2022) for a more detailed discussion of the reasoning behind the uncertainty assessment and the limitations of this approach, both of which can also be applied to our data set. A particular point to bear in mind is that stated values for the various uncertainty components are estimated based on previous studies or experience in the field rather than quantitatively determined.

Due to the temporal extrapolation required to derive fixed date values (annual and winter mass balance) from intermediate measurements, uncertainties for the fixed date values are higher than for the intermediate readings. This is reflected predominantly in the measurement quality flag and associated uncertainties, as well as in the density quality flag and density uncertainty for the stake data. This is a "one-size-fits-all" approach that does not account for the size of the temporal gap between the fixed date and the closest floating date, nor for the meteorological conditions affecting ablation and/or accumulation for the duration of said gap. We consider the resulting uncertainty a first order estimate useful for a comparison of the fixed date values with the intermediate data. A more detailed assessment of the uncertainties introduced by the temporal extrapolation is needed for future work.

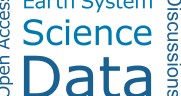

| Glacier | Total number of intermediate measurements (stakes and snow pits, yearly average in parentheses) | Total number of annual point mass balance (stakes and pits, fixed date, yearly average in brackets) | Winter point mass balance (snow pits, total number and yearly average in parentheses. Fixed date) | Spring probing (fixed date) | Fall probing (fixed date) |
|---|---|---|---|---|---|
| MWK (2007-2022) | 947 (59.2) | 263 (16.4) | 61 (3.8) | 1496 | 433 |
| VK (2012-2022) | 546 (49.6) | 165 (15.0) | 32 (2.9) | 627 | 428 |

**Table 1.** Number of total intermediate and annual ablation stake and snow pit measurements available for MWK and VK, as well as yearly averages for intermediate, annual and winter point mass balance measurements and number of probing locations.

**Figure 2.** Monitoring network at MWK, overview: Glacier outlines since 1969 and locations of stakes (circle markers) and snow pits (square markers) for which annual point mass balance data is available, color coded by years.

### 2.2.2 Glacier area

Tables 2 and 3 list years for which glacier outlines from MWK and VK are available as well as the underlying data sources and references. For MWK, previously unpublished outlines for 2012, 2015, 2018, and 2022 mapped from orthophotos are made available as part of this publication. For VK, previously unpublished outlines for 2012 and 2018 are made available. All outlines generated as part of the monitoring programs were mapped to be consistent with GI3 (Fischer et al., 2015a). See Figures 2 and 3 for overview maps of the outlines as well as of the stake and snow pit monitoring networks.

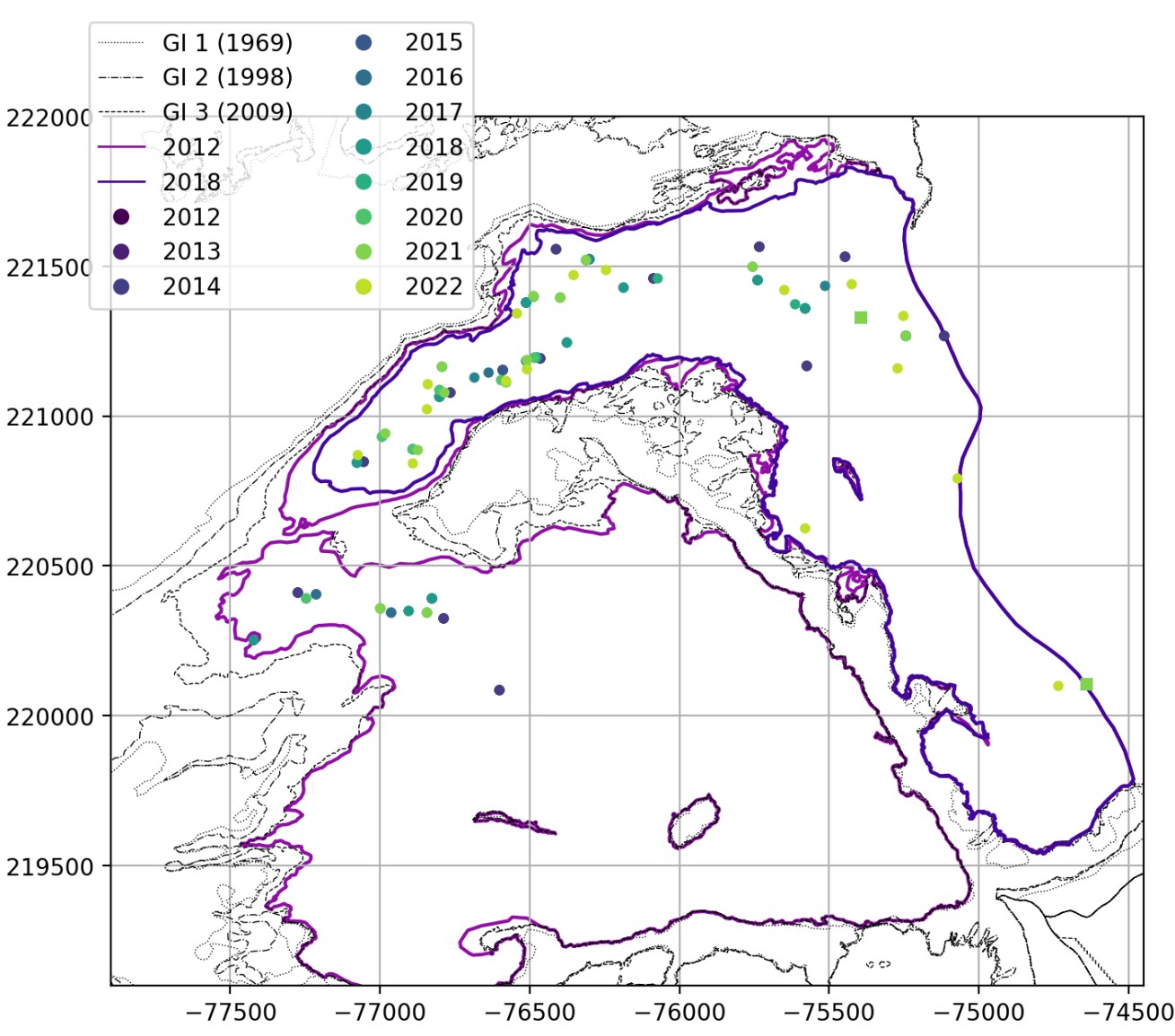

**Figure 3.** Monitoring network at VK, overview: Glacier outlines since 1969 and locations of stakes (circle markers) and snow pits (square markers) for which annual point mass balance data is available, color coded by years. Note that for 2018, only the VK outline is shown as Sulzbacherkees was not mapped.





| Year | Data type | References / source |
|---|---|---|
| 1850 (outline) | Little ice age inventory (GI LIA) | Fischer et al. (2015b); Groß and Patzelt (2015) |
| 1969 (outline) | First national glacier inventory (GI1) | Patzelt (1980); Groß (1987); Patzelt (2013) |
| 1998 (outline) | Second national glacier inventory (GI2) | Eder et al. (2000); Lambrecht and Kuhn (2007); Kuhn et al. (2012, 2013) |
| 2008* (DEM) | Airborne Lidar | Land Tirol (data.gv.at) |
| 2009* (outline) | Third national glacier inventory (GI3) | Fischer et al. (2015a) |
| 2012* (outline) | Orthophoto | orthophoto: Land Tirol (data.gv.at), outline: see Sec. 5 - data availability |
| 2015* (outline) | Orthophoto | orthophoto: Land Tirol (data.gv.at), outline: see Sec. 5 - data availability |
| 2015 (outline) | Fourth national glacier inventory (GI4) | Buckel and Otto (2018) |
| 2018* (outline) | Orthophoto | orthophoto: Land Tirol (data.gv.at), outline: see Sec. 5 - data availability |
| 2022* (outline) | Orthophoto | orthophoto: Land Tirol (data.gv.at), outline: see Sec. 5 - data availability |

**Table 2.** Glacier outlines and DEM for MWK with respective data sources. * marks data used for spatial extrapolation of glacier wide and elevation zone mass balance from point data.

| Year | Data type | References / source |
|---|---|---|
| 1850 (outline) | Little ice age inventory (GI LIA) | Fischer et al. (2015b); Groß and Patzelt (2015) |
| 1969 (outline) | First national glacier inventory (GI1) | Patzelt (1980); Groß (1987); Patzelt (2013) |
| 1998 (outline) | Second national glacier inventory (GI2) | Eder et al. (2000); Lambrecht and Kuhn (2007); Kuhn et al. (2012, 2013) |
| 2009* (outline) | Third national glacier inventory (GI3) | Fischer et al. (2015a) |
| 2012* (outline, DEM) | Airborne Lidar | DEM: Land Salzburg (data.gv.at), outline: see Sec. 5 - data availability |
| 2015 (outline) | Fourth national glacier inventory (GI4) | Buckel and Otto (2018) |
| 2018* (outline) | Orthophoto | orthophoto: Land Salzburg (data.gv.at), outline: see Sec. 5 - data availability |

**Table 3.** Glacier outlines and DEM for VK with respective data sources. * marks data used for spatial extrapolation of glacier wide and elevation zone mass balance from point data.





### 2.2.3 Glacier wide mass balance

Glacier wide mass balance was derived from the point measurements based on manual interpolation between the fixed date annual point measurements and extrapolation to unmeasured areas. The mapping of contour lines with equal mass balance (Østrem and Brugman, 1991) generally assumes linear gradients between measurement points and is additionally informed by expert knowledge of small scale topography as well as photographs and satellite imagery that show snow melt patterns. At VK, three automatic cameras take pictures of different sectors of the glacier multiple times per day. The imagery is used to assess snow line and ablation progression and contributes to the spatial extrapolation of mass balance to the glacier scale.

The elevation of the glacier terminus and the elevation zones used for the glacier wide mass balance product refer to the 2008 DEM for MWK and the 2012 DEM for VK, and the most recent glacier outline that was available in a given measurement year (Tables 2 and 3)

At VK, the 2009 outline from the third national inventory (GI3, Fischer et al. (2015a)) was adapted to reflect the separation of VK and SK and used for the mass balance until 2014. From 2014 to 2018, a new outline generated from the 2012 DEM was in use. The most recent VK outline is based on a high resolution orthophoto from 2018.

At MWK, the GI2 outline was adapted based on photos and differential GPS measurements for the mass balance seasons 2007/08 - 2009/10. For 2011/12 -2014/15, the GI3 outline was used. Outlines for the years 2012, 2015, 2018, and 2022 were mapped based on orthophotos as they became available.

To derive mass balance per elevation zone, the vector maps of interpolated glacier wide mass balance were integrated over each elevation zone. The equilibrium line altitude (ELA) is defined as the zero crossing of the vertical (elevation zone) profile of the specific mass balance (Cogley et al., 2011). The accumulation and ablation area ($S_c$, $S_a$) are respectively defined by the line of 0 mm w.e. in the glacier wide mass balance map and the most recent glacier outline. The total and specific mass balance in the accumulation and ablation area ($B_c$, $B_a$, $b_a$, $b_c$) are similarly delineated based on the 0 mm w.e. contour line.

Spatially integrated glacier wide mass balance derived from point data is prone to systematic errors due to, e.g., tilt and floating of stakes, local accumulation from avalanches, undersampling or non-representative sampling of certain areas, errors associated with the extrapolation method, and more. These errors accumulate for cumulative mass balance time series (Thibert et al. (2008); Zemp et al. (2013) and references therein). Unlike automatic or semi-automatic methods for spatial integration of point mass balance, the contour line method allows incorporation of expert knowledge related to mass balance relevant processes such as regular avalanche input in certain areas, wind drift, or the influence of crevasses. On the other hand, the method is to some degree subjective and dependent on the interpretations and specific knowledge of the observer drawing the contours.

A quantitative uncertainty assessment of glacier wide mass balance and detection of potential biases related to systematic errors requires an independent data set, typically of geodetic mass balance (e.g., Huss et al. (2009); Zemp et al. (2013)). Only one high resolution DEM is available for the period of record of glaciological measurements at MWK and VK, respectively. Hence, the glaciological data sets have not been compared to or validated against geodetic mass balance. We acknowledge that





this is a limitation of the presented data set. Reanalysis of the glacier wide time series is planned as soon as suitable new DEMs
become available.

### 2.2.4 Glaciological parameters - data format and access

The monitoring programs at MWK and VK are ongoing and the related data publications are "living documents" that are added
to as new data becomes available. The glacier wide fixed date annual mass balance as well as corresponding elevation zone
data are available as tabular data via the World Glacier Monitoring Service fluctuations of glaciers (FOG) database (WGMS,
2023). New data is added with each new iteration of the FOG database published by the WGMS and typically has a temporal
lag of two hydrological seasons.

Additionally, multiple data publication series are maintained for MWK and VK on the PANGAEA data platform. In the
PANGAEA data ecosystem, publication series are "parent publications" that can have multiple associated "child" data sets,
which are each issued a separate doi linked to the "parent doi" of the publication series in a relational database. The doi of the
publication series leads to a landing page where all associated data sets are listed and linked to.

The initial PANGAEA publication series entries for MWK (Stocker-Waldhuber et al., 2016) and VK Seiser and Fischer
(2016) were created in 2016. Data for each new measurement season has since been added sequentially as single year tabular
data. Further publication series for MWK and VK are listed in table 7. Point mass balance data was uploaded to the respective
publication series in tabular format for the period of record (Seiser et al., 2024d). Additional data for coming years will be
added as new entries. Publication series are available for glacier wide mass balance, glacier outlines, point mass balance, and
meteorological data at MWK and VK, respectively. See section 5, Table 7 for references and doi numbers.

The data sets in the publication series are curated by the authors of the data publications and the PANGAEA editing team
to ensure consistent formatting and correct integration into the relational database of PANGAEA. The annual associated data
sets can be cited individually or as part of the publication series. For example, the mass balance maps of VK for 2021/22
are referred to on PANGAEA as "*Seiser, B. et al. (2024): Glacier mass balance of Venedigerkees, Austria, 2022. (2024)* In:
*Seiser, B.; et al. (2024): Glacier mass balance of Venedigerkees, 2011/12 et seq.*" This data set contains shapefiles of winter
and annual mass balance and elevation zones for the 2021/22 season.

We note that there are various possible approaches to the challenge of timely and user-friendly publication of citable data sets
from ongoing monitoring which produces a range of data products at annual and sub-annual frequencies. For MWK and VK,
the time series of glacier wide and elevation zone mass balance can be extracted from the WGMS FOG database as a single
table. The PANGAEA publication series approach allows for more flexibility when adding data sets as they become available
in varying formats and reduces the temporal lag of new data being published. Tabular publication series, e.g. of annual and
elevation zone mass balance, can be bulk-downloaded through the publication series landing page and then merged to obtain
a time series for the period of record. For users wanting direct access to tabular, aggregated summary data for the period of
record with the option to easily compare MWK or VK with different sites, we recommend access through the WGMS FOG
database. For users interested in a larger variety of data and data formats specifically for MWK and VK, we recommend the
respective PANGAEA publication series and associated data sets.



## 2.3 Meteorological parameters

Standard meteorological parameters (Tab. 5) are measured at two automatic weather stations (AWS) located in about 500 m
distance from MWK and 800 m from VK. Subsets of current data can be viewed online through the data portal Lawis.at, where
the stations are named "Defreggerhaus" and "Keeskogel Südgrat", respectively. For the sake of clarity, we use AWS_MWK and
AWS_VK in the following to refer to the AWS near MWK ("Defreggerhaus") and VK ("Keeskogel Südgrat"). AWS_MWK
was established on 2020-08-10 and is located at 2978 m, 47.0839° N, 12.3634° E. The AWS tower is mounted in a relatively
sheltered, flat area near Defreggerhaus, a mountain hut. AWS_MWK did not initially include radiation sensors. These were
added on 2021-09-21. AWS_VK was established on 2019-09-19 at 3002m, 47.1317°N, 12.3142°E. It is situated on a wind-
exposed rock outcrop. Sensor specifications and related information is provided in 4.

For basic quality control, the data were scanned for data gaps and error values outside of the given sensor range as well as
for battery voltage below critical values for the operation of the data logger. Additional processing steps and quality flags were
applied as follows. Tab. 5) indicates the percentage of missing or erroneous data identified for each parameter. Derived data
products as presented in this publication, such as aggregated values for daily and monthly statistics, were computed without
interpolation of gaps unless otherwise stated. Positive degree day sums (Cogley et al., 2011) were computed based on daily
mean temperatures. All times in this manuscript are given in UTC unless otherwise stated.

The meteorological records are available as raw, uncorrected data and in a quality controlled version with quality flags at
10 minute intervals as logged by the station (PANGAEA publication series: Stocker-Waldhuber et al. (2024b), Seiser et al.
(2024b)).

### 2.3.1 Wind speed and direction

To account for possible riming effects in the wind speed and direction measurements, data was flagged as erroneous if wind
speed was 0 for more than one hour, or if wind direction did not change within a defined threshold for three or more consecutive
measurements. The threshold was set to three degrees based on the sensor resolution given by the manufacturer. Wind speed
for gusts was not given an extra riming flag. The riming flags for the 10 minute averages can be applied to the gust values if
needed.

### 2.3.2 Radiation

Four component radiation is measured at both AWS (shortwave incoming and outgoing/reflected, longwave incoming and
outgoing). Long wave (LW) radiation is corrected with the sensor temperature so that:

$LW\_Cor = LW + \sigma * T_{sensor}^4$

Where LW_Cor is the corrected radiation value, LW is the measured 10 minute average of incoming or outgoing long wave
radiation, $T_{sensor}$ is the 10 minute average sensor temperature in Kelvin (logged as "NR01K_Avg" in the data files) and $\sigma$ is
the Stefan-Boltzmann constant. No quality flags were set for long wave radiation aside from basic outlier and gap detection.





Short wave (SW) radiation was flagged if the value for outgoing radiation is greater than the value for incoming radiation.
This predominantly occurs when the sun is below the horizon. Day-time occurrences are relatively rare but also present in the
dataset, particularly in winter. This might occur due to site-specific geometric or topographic effects (e.g., reflection effects or
sun shining into the down-facing sensor at low sun angles), hoar frost or rime on the sensor, or sensor drift. The total percentage
of flagged SW values is about 35% at AWS_MWK and about 50% at AWS_VK (Tab. 5). Considering only day-time hours,
around 6-7% of the data have flags for SW outgoing > SW incoming.

### 2.3.3 Snow height and precipitation

Both stations were equipped with ultrasonic sensors to measure snow height for a sub-period of the overall time series. The
distance value measured by the sensors is corrected with air temperature so that:

$$Dist\_Cor = Dist\_Avg * \sqrt{(Tair\_Avg + 273.15)/273.15}$$

And:

$$snow\_h = h\_s - Dist\_Cor$$

Where Dist_Cor is the corrected distance value in m, Dist_Avg is the 10 minute mean as logged at the station, Tair_Avg is
the 10 minute mean air temperature, h_s is the height of the sensor above ground and snow_h is the snow height. Negative
values were flagged as errors. Due to a substantial amount of noise and outliers in the data, any value where the difference to
the subsequent value is greater than two cm was flagged to indicate that it should be treated with caution. In a second filtering
step, the difference of each remaining value to the 48 hour running mean was computed and values more than 15 cm higher or
lower than the mean were additionally flagged. The negative-value flag accounts for most of the flagged percentage value given
in Table 5. The remaining flags make up about 5% of the total at AWS_VK and 15% at AWS_MWK, which has more frequent
occurrences of flags for "large difference between consecutive data points". A comparison of outlier-filtered snow depth data
and the complete, unfiltered data set is shown in Fig. S1 and S2 (supplementary material). At both stations, the ultrasound
sensors failed after approximately two seasons of operations, with some prior deterioration of data quality. The location of
AWS_VK is very wind exposed and snow rarely accumulates in the direct vicinity of the station. Accordingly, the snow height
data from AWS_VK is not representative of the surrounding area. AWS_MWK is in a more sheltered location where snow
accumulates and builds a seasonal snowpack.

At AWS_MWK, an unheated precipitation gauge is used to measure precipitation. This data is considered reliable only for
liquid precipitation and under the assumption that the gauge is not affected by rime or snow cover, i.e. for rain in the warmer
seasons. As a first order filter, precipitation data is flagged as unreliable when air temperature is between 0 and 4°C and as
erroneous when air temperature is below freezing. This results in almost 80% of data being flagged for low temperature criteria
(Table 5).



| Parameter | Sensor name and manufacturer | Specifications |
|---|---|---|
| Air temperature | Campbell Scientific Rotronic HC2S3 | Range: -40°C to +60°C Accuracy: ±0.1°C with standard config. |
| Relative humidity | Campbell Scientific Rotronic HC2S3 | Measurement range: 0 to 100% Accuracy: ±0.8% at 23°C |
| Precipitation (AWS_MWK) | Campbell Scientific 52203 Rain Gauge (unheated) | Resolution: 0.1 mm per tip Accuracy: 2% up to 25 mm/hr 3% up to 50 mm/hr |
| Snow height | SR50A Sonic ranging sensor | Range: 0.5-10 m Resolution: 0.25 mm Accuracy: ± 1 cm |
| Wind speed and direction | R.M. Young 05103-45 Wind Speed & Direction Sensor | Measurement range: speed 1-60 m/s \| dir 360 deg Accuracy: speed ±0.3 m/s \| dir ± 3 deg |
| Surface atmospheric pressure | CS106 Vaisala PTB110 | Range: 500 to 1100 hPa Accuracy: ±1.5 hPa at -40° to +60°C) |
| Radiation | Hukseflux NR01 Net Radiometer | Shortwave range: up to 2000 $W/m^2$, Spectral range: 305-2800 nm. Longwave range: up to 1000 $W/m^2$, Spectral range: 4500-50000 nm, Expected accuracy: ± 10% on daily sum. Calibration date AWS_MWK: 2018-09-03, Uncertainty AWS_MWK: $±0.25x10^-6V(W/m2)$, Calibration date AWS_VK: 2017-05-23, Uncertainty AWS_VK $±0.26x10^-6V(W/m2)$ |

**Table 4.** Sensor specifications for AWS_MWK and AWS_VK. Data logger at both stations: Campbell Scientific CR3000.

# 3 Results and discussion

## 3.1 Glaciological parameters

### 3.1.1 Intermediate measurements

The intermediate data sets contain stake readings throughout the summer seasons and snow pit data for the spring and fall surveys. The VK data set contains readings for three ablation stakes on neighboring SK in addition to the stakes on VK. In total, the VK intermediate data set contains measurements for 32 spring snow pits, 17 fall snow pits, and 497 intermediate stake readings. The MWK intermediate data set contains 61 spring snow pit measurements, 37 fall pits, and 849 stake readings.

The median number of days between the spring survey and the fixed date (April 30) is 11 days at VK and 6 at MWK. In fall, the median difference is 5 days at VK and 8 days at MWK. Ablation stake readings typically begin in late June or July once the seasonal snow cover has mostly melted and are carried out in irregular intervals depending on conditions (Fig. 4, panel a).





| Parameter and station | Start | End | percentage flagged (% of total measurements) |
|---|---|---|---|
| Air temperature MWK | 2020-08-10 | 2023-10-31 | 0.0 |
| Air temperature VK | 2019-09-19 | 2023-10-31 | 0.0 |
| Relative humidity MWK | 2020-08-10 | 2023-10-31 | 0.0 |
| Relative humidity VK | 2019-09-19 | 2023-10-31 | 0.0 |
| Precipitation MWK | 2020-08-10 | 2023-10-31 | 79 (low temperature flags) |
| Snow height MWK | 2019-07-01 | 2022-06-02 | 41 |
| Snow height VK | 2020-07-01 | 2022-06-28 | 32 |
| Wind speed and direction MWK | 2020-08-10 | 2023-10-31 | Wind speed: 0.9, wind direction: 3.3 |
| Wind speed and direction VK | 2019-09-19 | 2023-10-31 | Wind speed: 2.0, wind direction: 3.1 |
| Surface atmospheric pressure MWK | 2020-08-10 | 2023-10-31 | 0.0 |
| Surface atmospheric pressure VK | 2019-09-19 | 2023-10-31 | 0.0 |
| Shortwave radiation MWK | 2021-09-21 | 2023-10-31 | SWin: 34.7 SWout: 35.1 |
| Shortwave radiation VK | 2019-09-19 | 2023-10-31 | SWin: 50.8 SWout: 51.5 |
| Longwave radiation MWK | 2021-09-21 | 2023-10-31 | LWin: 0.0 LWout: 0.0 |
| Longwave radiation VK | 2019-09-19 | 2023-10-31 | LWin: 0.1 LWout: 0.1 |

**Table 5.** Period of record and percentage of data flagged as erroneous for each parameter.

The corrections applied to floating date spring and end-of-season surveys to arrive at corresponding fixed date values have a
range of roughly -600 to +400 mm w.e. and correlate broadly with the number of days between the floating and the fixed dates
(Fig. 4, panel b). The mass balance error associated with the different data types is lowest for the intermediate readings of stake
data (Fig. 4, panel c). Compared to the stake readings, intermediate snow pit data have a higher uncertainty associated with
the method of the measurement as well as more variation in density and higher density uncertainty. Uncertainty values for the
intermediate measurements are a similar range as those found by Geibel et al. (2022).

Figure 5 shows cumulative ablation at selected stakes to illustrate the subseasonal resolution of the intermediate stake data.
The figure highlights the range of ablation rates and the length of the ablation season at different stakes, as well as interannual
variations at each stake. 2022 stands out as an exceptional year particularly at higher elevations. VK stake 95 (Fig. 5, panel a)
is a relatively low elevation stake on the tongue of VK (elevation VK stake 95: 2583 m-2595 m). Here, the overall ice loss of
summer 2022 and the seasonal progression is comparable to 2015, another very negative year. VK stake 100 (Fig. 5, panel b)
and MWK stake 11 (Fig. 5, panel c) are at higher elevations (VK stake 100: 2739 m-2876 m; MWK stake 11: 3027 m - 3055
m) and ablation begins one to two months later than at VK 100 in most years of the time series. At these stakes, 2022 was
unprecedented in the time series in terms of the total ablation as well as the length of the ablation season.

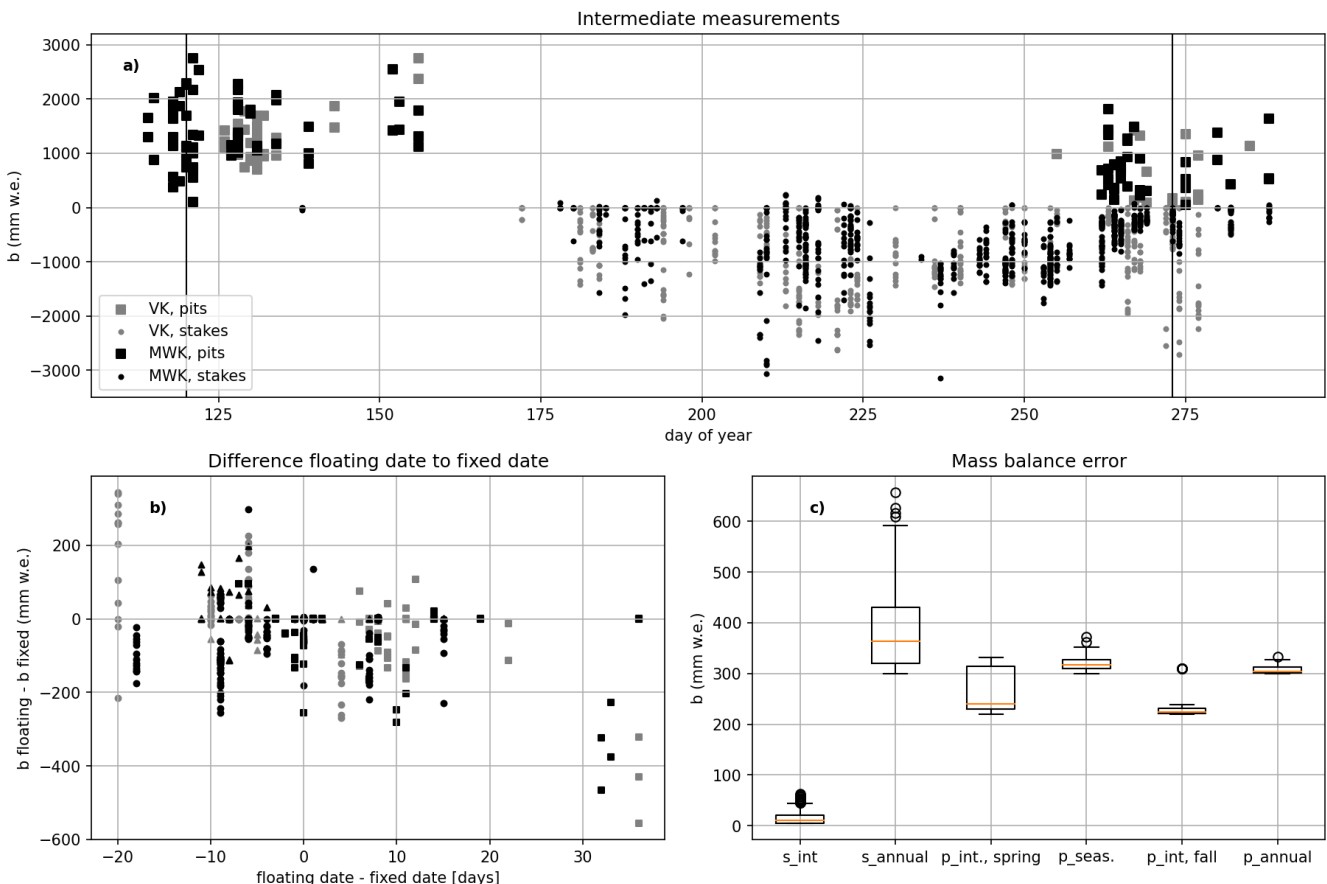

**Figure 4.** a):Intermediate mass balance measurements (pits and stakes) at both sites plotted against the day of the year (all years of the time series). The black vertical lines indicate April 30 (day of year 120) and September 30 (day of year 273) in non-leap years. Positive stake readings were recorded in some cases at MWK to indicate the formation of superimposed ice at the stake. Zero values indicate that the stake was snow covered during the measurement period and no ablation took place. b): Difference of floating date to fixed date mb values for the spring and fall snow pits and the stake readings closest to the fall fixed dates. Positive values on the x-axis mean that the floating date measurement took place after the corresponding fixed date. Positive values on the y-axis mean that b (floating date) > b (fixed date). c): Mass balance error for intermediate stake readings (s_int), annual fixed date stake values (s_annual), floating date spring snow pit data (p_int, spring), fixed date winter values at the snow pit locations (p_seas), floating date fall snow pit data (p_int, fall), fixed date annual values at the snow pit locations (p_annual).

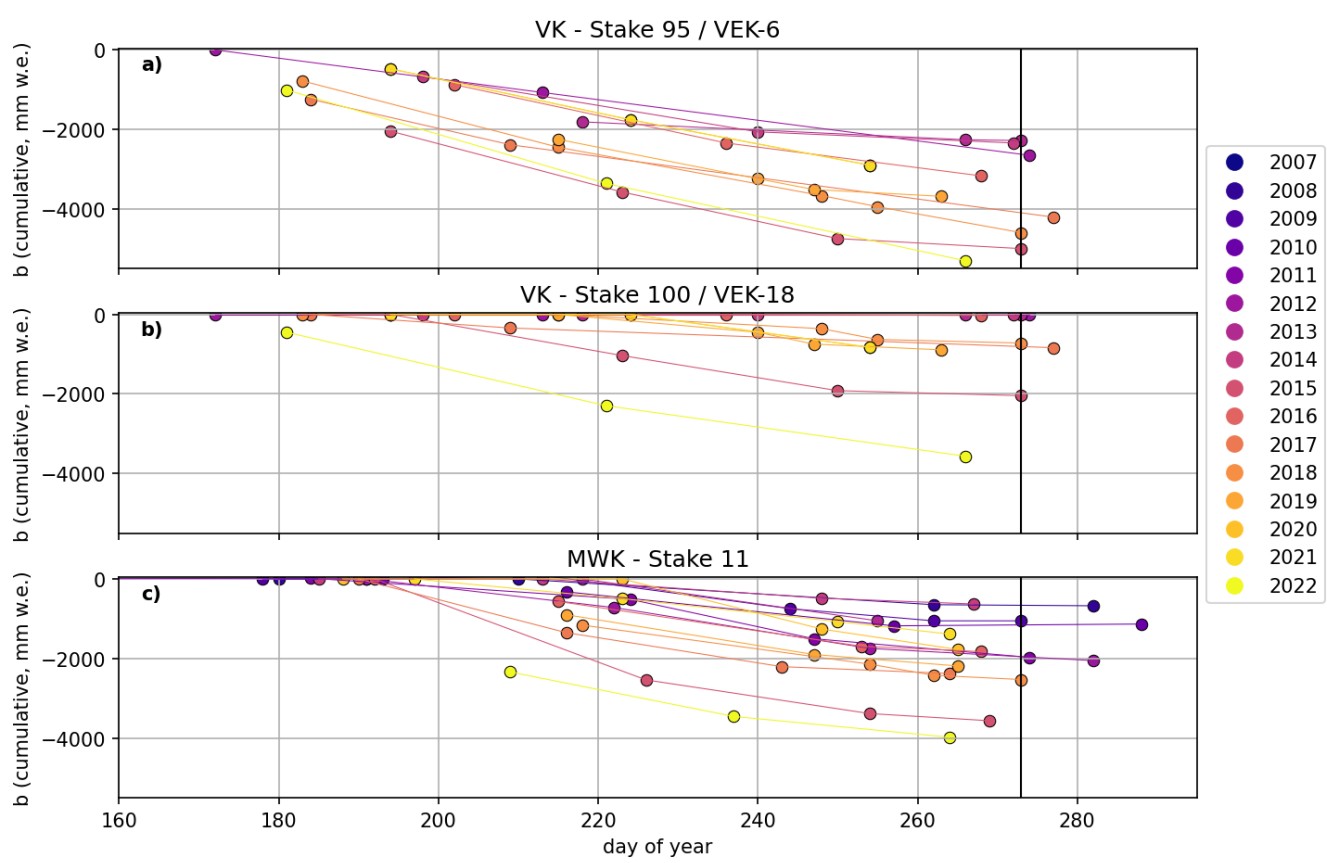

**Figure 5.** Intermediate cumulative mass balance at three selected stakes plotted against the day of year of the respective measurements. Minimum (maximum) elevation of VK Stake 95 (panel a, later renamed to VEK-6) throughout the time series: 2583m (2595m). Minimum (maximum) elevation of VK Stake 100 (panel b, later renamed to VEK-18): 2739m (2876m). Minimum (maximum) elevation of MWK Stake 11 (panel c): 3027m (3055m).



### 3.1.2 Annual and winter point mass balance and spatial integration

Counting ablation stakes and fall snow pits, the number of annual fixed date point mass balance data points varies between
12 and 20 at MWK and 13 and 17 at VK (including the stakes on SK). The fixed date data have higher mass balance errors
than the intermediate measurements due to the added uncertainties in the density and measurement quality components of eq.
1 and eq. 2. The density uncertainty is given relative to mass balance, hence the overall mass balance error is higher when the
absolute magnitude of b is large. The fixed date point mass balance data has higher mass balance errors than the intermediate
data. The mass balance error is typically in the range of the $\pm30$ cm "measurement quality" error added to account for the
temporal extrapolation (Figure 4, panel c).

The most negative annual point mass balance of the VK time series was recorded at an elevation of 2536 m in 2022 with
-5841 $\pm$ 657 mm w.e. At MWK, the most negative value of the time series was measured at 2728 m in 2012 with -4878 $\pm$573
mm w.e. The respective ablation stake was no longer operational during the very negative summer of 2022 due to loss of glacier
area at this elevation. In 2022, the lowest ablation stake of the MWK monitoring network was located at 2807 m and recorded
an annual point mass balance of -4533 $\pm$544 mm w.e. Figure 6 shows winter (panels a, b) and annual (panels c, d) fixed date
point mass balance for all ablation stakes and snow pits.

Accumulation is typically measured at three snow pit locations at VK and four at MWK. 2022 was the first year in which
no snow remained in fall at the snow pit locations at either glacier and, hence, no accumulation was measured. Winter mass
balance tends to increase with elevation at VK (Fig. 6, panel b), while MWK (Fig. 6, panel a) has a non-linear mass balance
gradient with maximum accumulation slightly below the summit region. This curve in the gradient is less pronounced but also
evident in the annual point mass balance data (Fig. 6, panel c).

In addition to fixed date stake and snow pit point mass balance, Figure 6 shows depth probing data within $\pm100$ mm w.e.
of the spatially integrated mass balance value for the given elevation zone for the spring and annual surveys, respectively.
This represents only a relatively small fraction of the total probe points in the data sets, highlighting the challenges of the
method. It is difficult to assess uncertainties and errors of the snow probing data without additional information on conditions
during the measurement campaigns. Values flagged as not being in agreement with the annual elevation zone mass balance
are often measurements of fresh snow depth and, hence, do not represent annual mass balance. This data is used as additional
information informing the extrapolation from the point scale to elevation zones and glacier wide mass balance. For the winter
mass balance, snow depth probing aims to identify the previous year's firn horizon. Definitive identification of this horizon
can be hampered by, e.g., ice lenses in the snow pack. We include the probing data sets for the sake of completeness and
transparency and encourage any potential users of this data to interpret it with caution.

Glacier wide mass balance was derived from a minimum (maximum) of 12 (19) fixed date point mass balance values at
MWK and 13 (17) at VK, respectively. At VK, annual mass balance increases monotonically with elevation in most years, with
some variation of patterns in the accumulation zone near the ice divide to Untersulzbachkees. As with the point mass balance
data, 2022 stands out as an extremely negative year. Figures 7 and 8 show the manually drawn contour lines of mass balance
and the resulting spatial pattern of fixed date annual and winter mass balance, as well as the fixed date point mass balance





data for each year and glacier. The magnitude of ablation in 2022 as well as the spatial patterns in the lower section of the glacier were broadly comparable to 2015 and 2018, while ablation in the upper sections (previously the accumulation area) was unprecedented in the period of record.

At MWK, the spatial patterns in the contour line maps (Fig. 9, 10) provide additional insights into the curved mass balance gradient apparent in the point-scale data. Accumulation is typically greatest in an elevation band slightly below the highest sector of the glacier. Within this elevation band there are two distinct zones that tend to show the highest winter and annual accumulation (in 2022: the least ablation). These patterns are related to wind deposition and erosion of snow. Like at VK, the ablation in the upper regions of MWK in 2022 was unprecedented for the time series.

Comparing point mass balance data with the extrapolated mass balance map derived from the contour lines, it is apparent that point values generally match the spatially integrated values but there are exceptions in both directions (point value > spatially extrapolated value or point value < spatially extrapolated value). At VK, two to three stakes in the mid-section of the tongue have less negative values than the spatially integrated value e.g. in 2015, 2019, and 2020. This is related to reduced ablation due to snow input from avalanches at these stake locations and constitutes a known bias that is manually corrected for when
drawing the contour lines.



**Figure 6.** Winter (top) annual (bottom) point mass balance vs elevation of the measurement point, color coded by year for MWK (panels a, c) and VK (panels b, d). The mass balance error for stakes and snow pits is shown as thin black lines. Small circles show the probe data used to inform the spatial integration of the point mass balance data to the glacier scale both for the winter and annual mass balance. Errors bars for the probing points not plotted for the sake of readability. The probing error is given in the data file and is computed from a 30 cm quality error for the fixed date interpolation, the type error, and the density error.

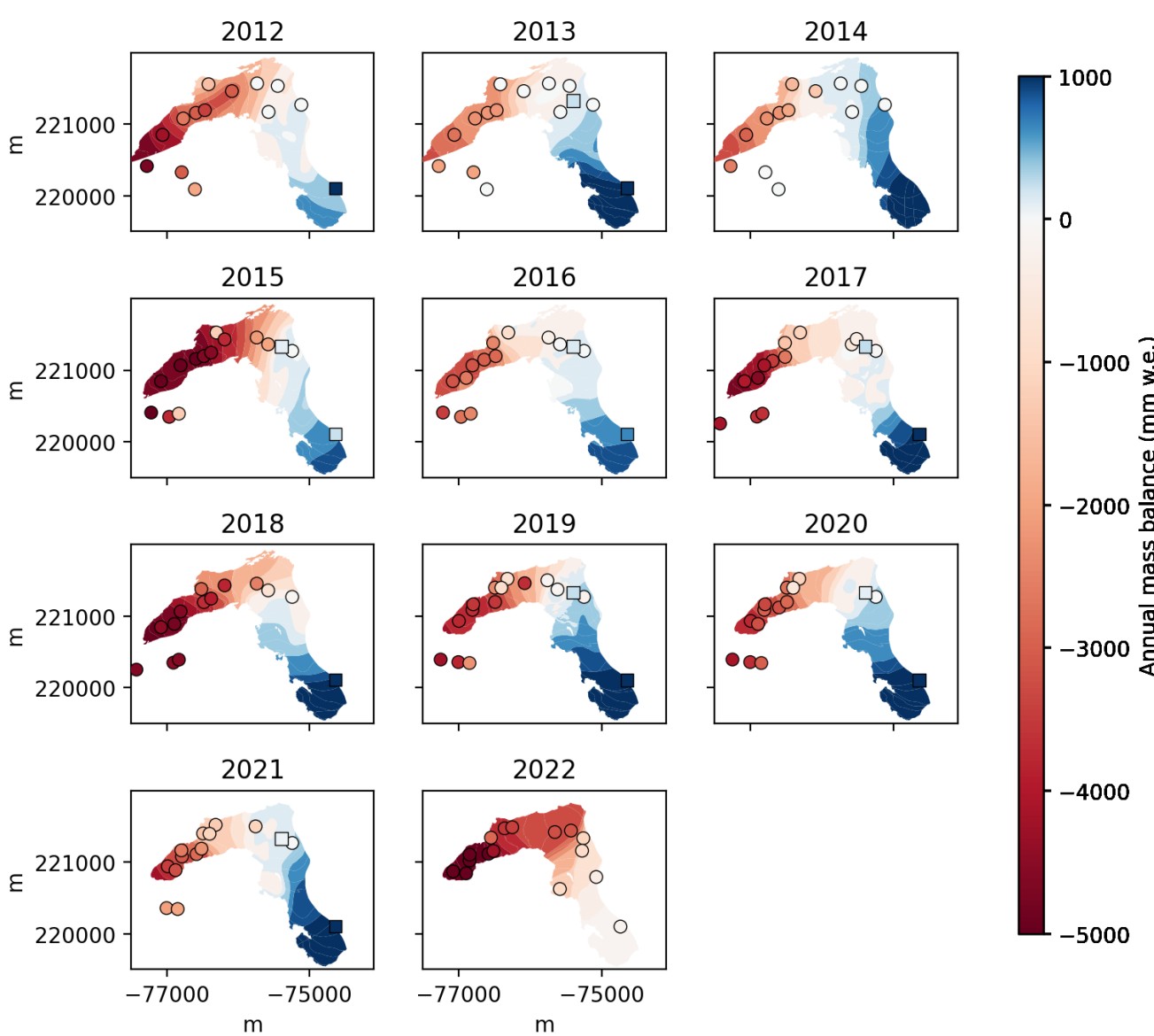

**Figure 7.** Annual fixed date point and elevation zone mass balance at VK. Circles: abaltion stakes. Squares: snow pits. Coordinate reference system: MGI / Austria GK Central (EPSG:31255).

**Figure 8.** Winter fixed date point and elevation zone mass balance at VK. Squares: snow pits. Small circles: snow depth probing points. Coordinate reference system: MGI / Austria GK Central (EPSG:31255).



**Figure 9.** Annual fixed date point and elevation zone mass balance at MWK. Circles: ablation stakes. Squares: snow pits. Coordinate reference system: MGI / Austria GK Central (EPSG:31255).



**Figure 10.** Winter fixed date point and elevation zone mass balance at MWK. Squares: snow pits. Small circles: snow depth probing points. Coordinate reference system: MGI / Austria GK Central (EPSG:31255).



### 3.1.3 Glacier wide annual and winter mass balance

The cumulative specific mass balance of MWK over the period of record (2007-2022) is -14.68 m w.e. (Fig. 11, panel c) This amounts to about 16 m of ice loss in total or about one meter of ice loss per year averaged over the entire glacier area. A slightly positive glacier wide annual mass balance was recorded in 2013/14 (b: 117 mm w.e., Fig. 11, panel a). All other years of the time series were characterized by mass loss. The most negative season to date was 2021/22 with a specific mass balance of -2449 mm w.e.

The shorter time series of VK (2012-2022) shows a cumulative specific mass balance of -8.79 m w.e., i.e. not quite 10 m of ice loss (Fig. 11, panel c). All years of the time series had a negative annual mass balance (Fig. 11, panel b). The 2013/14 season was the least negative of the time series with b = -152 mm w.e. Like at MWK, the greatest losses were recorded in 2021/22 with b = -2209 mm w.e.

The standard deviation of b is 645 and 639 mm w.e. at MWK and VK, respectively. The annual variability of the winter mass balance is considerably lower at both sites with a standard deviation of 292 and 171 mm w.e., respectively. At MWK, 2006/07 had the lowest specific winter mass balance ($b_w$) of the time series with 674 mm w.e. followed by 2021/22 with 816 mm w.e. At VK, $b_w$ was lowest in 2015/16 with 1116 mm w.e. The largest winter accumulation was recorded in 2018/19 at MWK ($b_w$ = 1751 mm w.e.) and in 2016/17 at VK ($b_w$ = 1735 mm w.e.).

Fig. 12 shows seasonal and annual mass balance by elevation zones for both sites. At VK, the highest elevation zones generally see the greatest winter accumulation and least summer and annual ablation, as in the point data and the contour line maps. In contrast, at MWK the greatest winter snow accumulation is usually found about 200-300m below the highest sections of the glacier due to local wind drift and deposition patterns. Annual and summer ablation show similarly curved gradients with higher loss rates in the uppermost elevation zone (3400-3450) than between approximately 3100 and 3200 m.
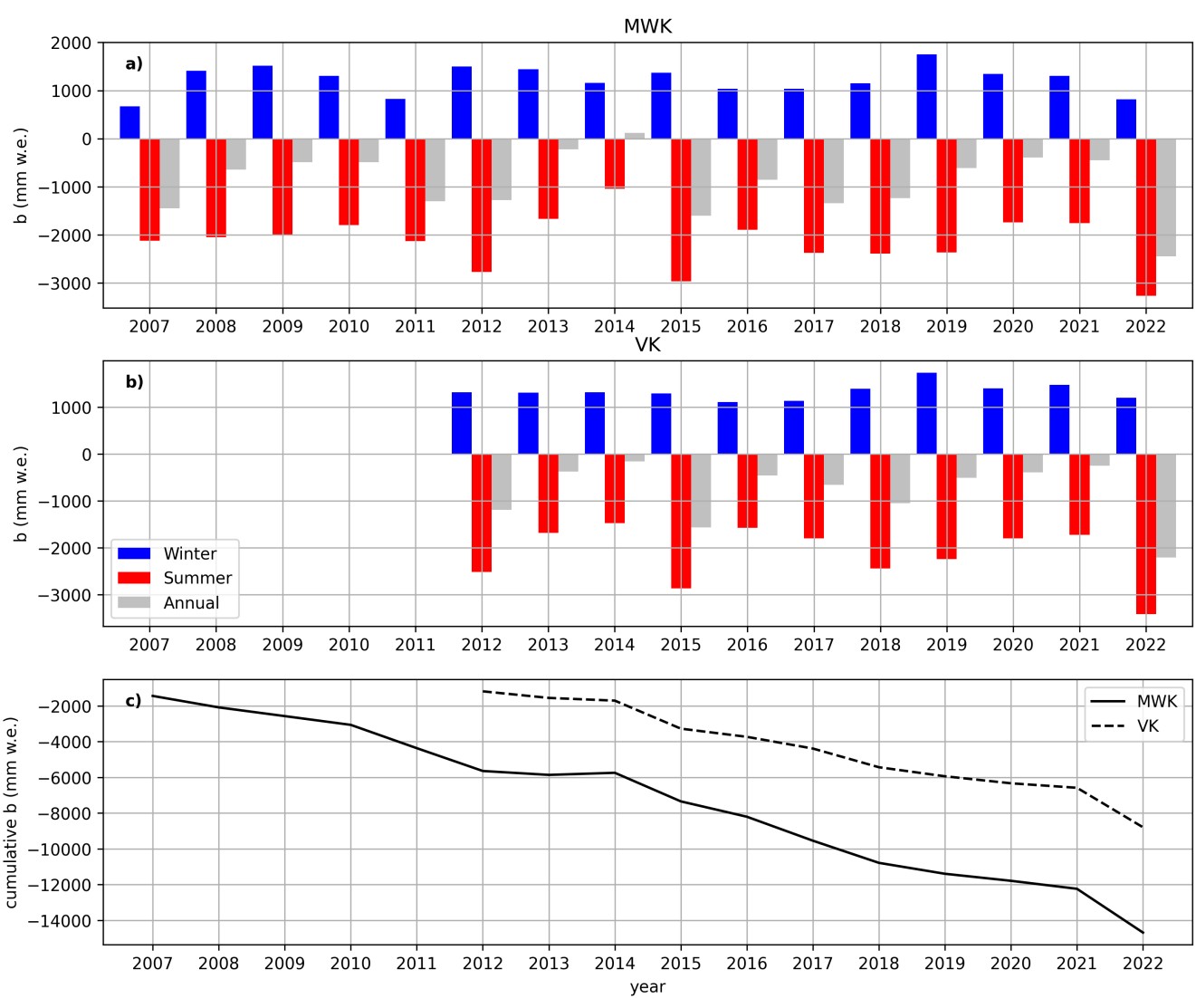

**Figure 11.** Seasonal (winter, summer) and annual glacier wide specific mass balance at MWK (panel a) and VK (panel b). Panel c shows cumulative specific mass balance at MWK and VK.

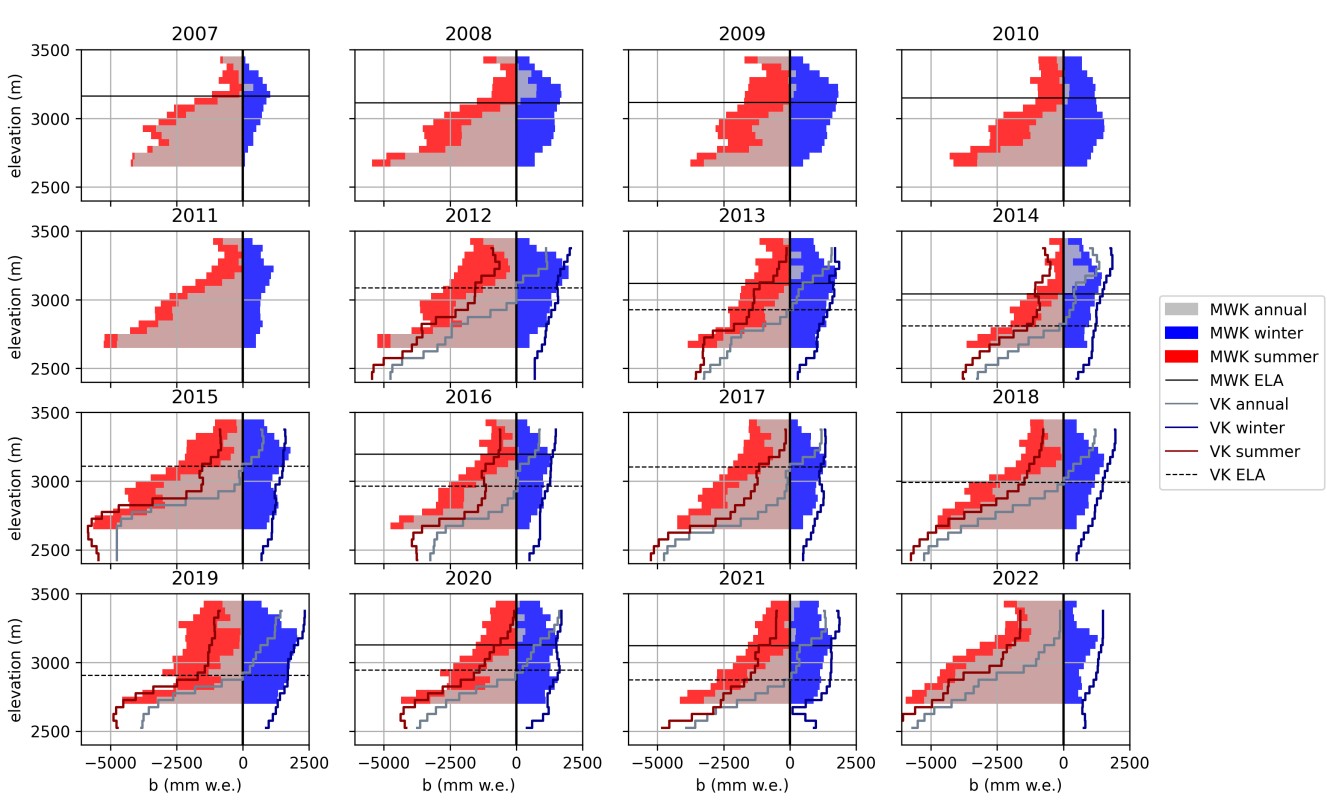

**Figure 12.** Annual and seasonal fixed date specific mass balance per 50m elevation zones at MWK (2007-2022) and VK (2012-2022). Horizontal lines indicate the equilibrium line altitude (ELA; solid: MWK, dashed: VK). ELA is not plotted when it is above the summit elevation.





## 3.2 Meteorological parameters

The AWS datasets show seasonal and diurnal variations of temperature, humidity, radiation, and other measured parameters in line with expectations for the station locations and elevations (Fig. 13 and Fig. S3 in the supplement). Table 6 lists monthly mean temperatures at both stations. January and December are the coldest months at both sites. July and August are warmest on average. Mean daily high temperatures in July are 7.4°C at AWS_MWK and 5.8°C at AWS_VK. The mean daily temperature amplitude is also highest in the summer months, reaching 3.8°C in June at AWS_MWK and 2.3°C in July and August at AWS_VK. Considering monthly mean temperatures, September 2023 stands out as exceptionally warm. September was the warmest month of the year in 2023. The all-time highest and lowest temperatures measured at AWS_MWK were 15.9°C on 2023-08-22 (13:20) and -23.82°C on 2021-02-14 (06:30), respectively. At AWS_VK, the corresponding values are 15.02 °C on 2023-09-06 (11:40) and -23.86°C on 2022-12-11 (20:40).

Considering cumulative positive degree day (PDD) sums (Fig.14) for years in which data is available from the beginning of the fixed date ablation season (AWS_VK: 2020-2023; AWS_MWK: 2021-2023), it is apparent that PDD sums began cumulating earlier in 2022 than in 2021 and 2023. The 2023 PDD season began with similar increase rates as 2021 but picked up considerably in mid-August and surpassed the 2022 sums by the end of the ablation season.

Wind blows predominantly from SSW or NNE at AWS_MWK, with higher wind speeds typically occurring in NNE cases. At AWS_VK, the main wind direction is NW to W (Fig. S4 supplementary material). The highest 10 minute gust speed of the station record was 28.02 m/s (direction: 32.6°) on 2022-01-05 at AWS_MWK and 34.55 m/s (292.9°) on 2020-02-03 at AWS_VK, respectively. At AWS_MWK, the highest 10 minute average wind (23.96 m/s) speed occurred at the same time as the gust record. At AWS_VK, the highest 10 minute average was recorded on 2020-02-27 with 26.17 m/s.

Snow height measurements were hampered by sensor failures (Figs. S1, S1 supplementary material) and are available only for the winter season of 2020-21 and 2021-22. In 2020-21, the ground at AWS_MWK (Fig. 13) was snow-covered from about October to June, with maximum snow heights of close to two meters in May. Snow melt began in early June and proceeded rapidly. The 2021-22 snow season was very dry in comparison to 2020-21. The snowpack began building in October but never consistently grew to more than about 0.5 m. Snow depth dropped to near zero in March before late spring snow falls brought a renewed increase. The 2021-22 snow season ended in late May.

Fig. 13 shows monthly precipitation sums as recorded by the unheated rain gauge at AWS_MWK when air temperature was > than 4 °C, i.e. likely to have been rain. August 2021 stands out as the wettest month of the station time series by a large margin, with a monthly sum of almost 400 mm. Most "warm" precipitation as defined via the aforementioned temperature threshold occurs between July and October, with considerable interannual variation in the length of the "warm precipitation season".

The precipitation and snow depth data clearly have limitations related to the characteristics of the instrumentation (unheated rain gauge, snow height sensor problems and wind exposed location at VK) and need to be carefully evaluated for any further use. We present the raw data as well as a version with preliminary quality control flags to allow application specific adaptation

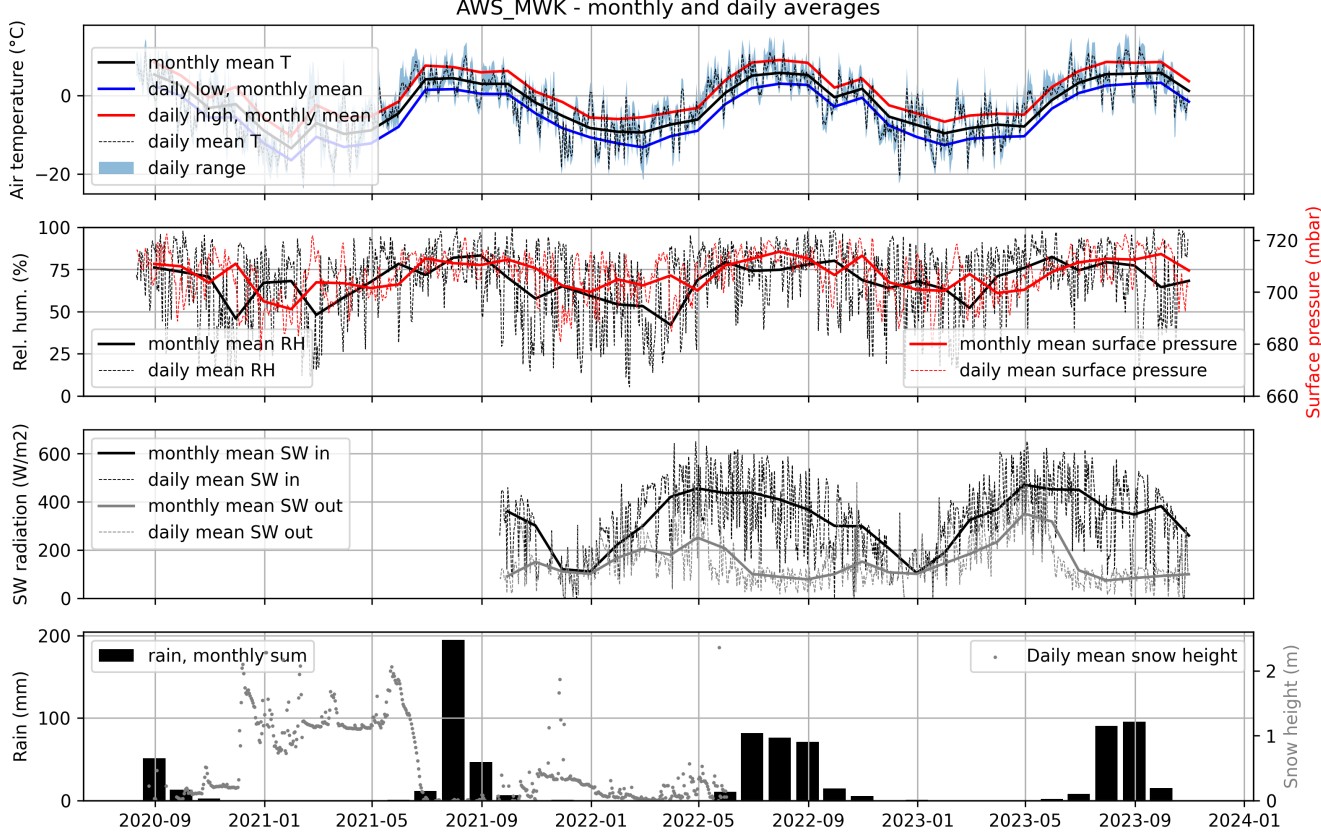

**Figure 13.** Daily and monthly averages for air temperature, relative humidity, surface air pressure, shortwave incoming and outgoing radiation, snow depth and rain (unheated precipitation gauge) for the period of record at AWS_MWK. See Fig. S3 in the supplementary material for an analogous figure for AWS_VK.

as needed. Documentation of future sensor changes and additional data will be published with the current data sets as it becomes
available.

In addition to systematic errors and uncertainties inherent to the sensors and measurement system, local and site specific weather phenomena should be kept in mind in further analyses. For example, wind direction as recorded at AWS_MWK shows a SSW-NNE dipole (Fig. S4 supplementary material). Southerly wind directions at the station can occur during larger scale northerly flow due to the formation of a lee rotor that affects conditions at MWK and AWS_MWK. At VK, wind speed and
direction are also strongly affected by the surrounding topography, as is common for AWS in mountainous terrain.

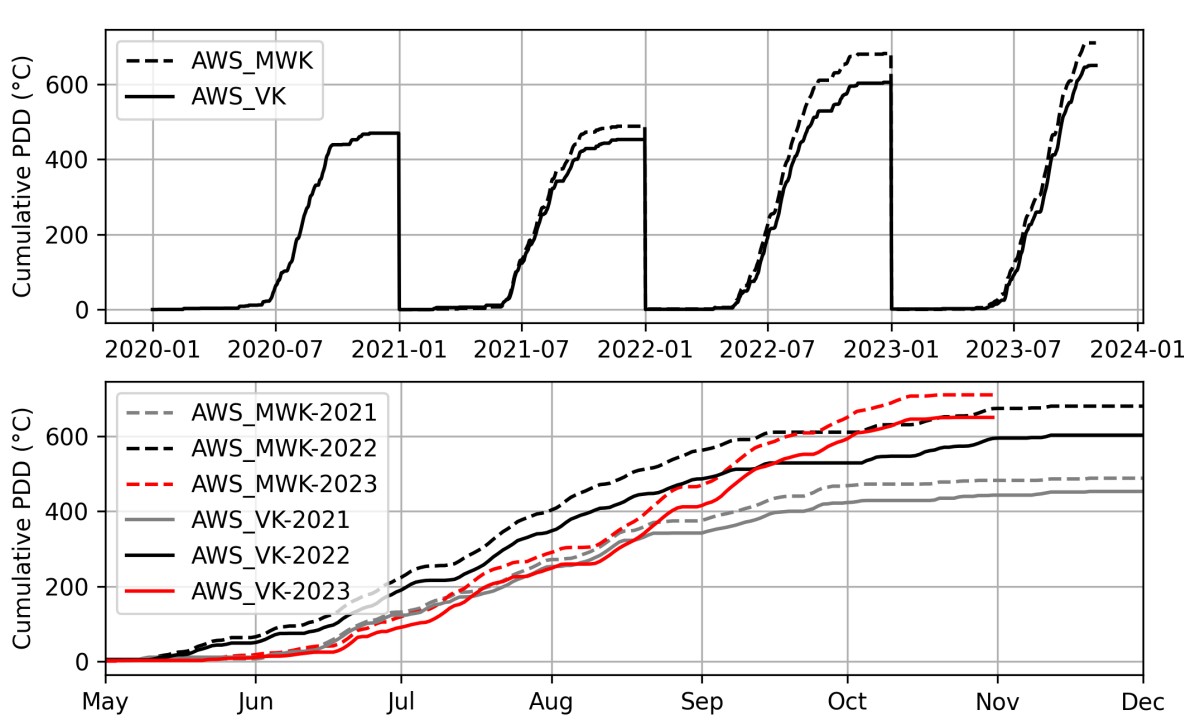

**Figure 14.** Top panel: Time series of annual cumulative positive degree day (PDD) sums at AWS_MWK and AWS_VK. Bottom panel: Comparison of cumulative PDD for 2021, 2022, 2023. AWS_MWK.



| month | 2019 | 2020 | 2021 | 2022 | 2023 | 2020 | 2021 | 2022 | 2023 |
|---|---|---|---|---|---|---|---|---|---|
| | VK | | | | | MWK | | | |
| 1 | / | -6.4 | -13.5 | -10.0 | -9.9 | / | -13.5 | -9.2 | -9.6 |
| 2 | / | -8.1 | -6.9 | -9.8 | -8.8 | / | -7.0 | -9.4 | -8.3 |
| 3 | / | -9.2 | -10.4 | -7.7 | -7.6 | / | -9.8 | -7.4 | -7.4 |
| 4 | / | -3.5 | -9.6 | -6.8 | -8.4 | / | -8.9 | -6.1 | -7.9 |
| 5 | / | -2.8 | -4.9 | -0.1 | -1.9 | / | -4.5 | 0.7 | -1.2 |
| 6 | / | 0.9 | 3.7 | 4.4 | 2.5 | / | 4.1 | 5.1 | 3.3 |
| 7 | / | 3.7 | 4.1 | 5.0 | 4.9 | / | 4.4 | 5.8 | 5.4 |
| 8 | / | 5.0 | 2.4 | 4.6 | 5.0 | 5.4 | 3.0 | 5.3 | 5.6 |
| 9 | / | 2.2 | 2.3 | -1.0 | 5.7 | 2.5 | 3.0 | -0.4 | 5.8 |
| 10 | 0.6 | -3.3 | -2.1 | 1.7 | 1.1 | -3.2 | -1.8 | 1.7 | 1.2 |
| 11 | -6.5 | -2.4 | -5.2 | -5.6 | / | -2.2 | -5.2 | -5.4 | / |
| 12 | -7.5 | -8.3 | -9.2 | -7.3 | / | -8.9 | -8.3 | -7.5 | / |

**Table 6.** Mean monthly temperatures at AWS_VK and AWS_MVK (°C).





## 4   Outlook and conclusions

Glacier change in the Alps is accelerating (Zemp et al., 2019; Sommer et al., 2020; Hugonnet et al., 2021). The monitoring networks at VK and MWK have already been adapted to account for some of the challenges this poses, mainly by extending the stake network into the former accumulation zones as firn area is lost. To ensure that the observing system can capture
processes of progressing deglaciation, further adaptations will be needed in the coming years. Moving forward, a key aspect is to more comprehensively account for changes to glacier area and elevation and to incorporate respective data into the mass balance monitoring system. To date, area change is mapped in approximately 3-6 year intervals and only one high resolution DEM is available per site. The frequency of glacier area mapping should be increased to capture the rapid emergence of rock outcrops in the thin upper section of MWK and the changing glacier tongue of VK. Multi-temporal DEMs are essential to
track elevation changes and a necessary requirement for a quantitative uncertainty assessment of glacier wide mass balance. Reanalysis of the mass balance time series based on calibration and validation with geodetic mass balance (Zemp et al., 2013) is urgently needed and will become increasingly important as potential existing inhomogeneities in the time series accumulate and new issues arise, e.g. due to rapid area change. Time series reanalysis and homogenization is planned as soon as new DEMs can be acquired.

Calibration of glaciological glacier wide mass balance with geodetic mass balance can address the biases and uncertainties inherent to spatial extrapolation from point mass balance. Temporal extrapolation from floating to fixed dates is another source of considerable uncertainty that is difficult to quantify. Recent years have shown that ablation at VK and MWK is not necessarily limited to May through September and can extend well into October. The fixed date system has historically been the standard for glacier mass balance monitoring in Austria. However, as climatic conditions continue to change, temporal
reanalysis of mass balance time series or adaptation of the measurement periods to a lengthened ablation season may become increasingly important to capture the actual seasonality of ablation and accumulation at the sites. Preserving intermediate mass balance measurements at the best available resolution along with key meteorological parameters is central to such efforts.

Sub-seasonal observations and modeling of mass balance help quantify short term fluctuations of runoff and associated changes in catchment hydrology and ecosystem impacts. The glacier monitoring programs of MWK and VK are located within
HTNP and complemented by additional long term monitoring and ecosystem research (Schütz and Füreder, 2018; Körner et al., 2020). The sites offer the possibility to observe ecological consequences of accelerated glacier retreat in a protected area, i.e. excluding direct anthropogenic influences by changing land use (Huemer, 2011).

The presented data expands upon the previously available time series of glacier wide mass balance at MWK and VK by adding annual and intermediate floating date point data. We second the points made by (Geibel et al., 2022) regarding the
importance of "rescuing" the frequently unpublished point data and comprehensive meta data documentation. The floating date point data is the basis for the spatial extrapolation to the glacier scale and, hence, a required component for future reanalysis and more detailed uncertainty assessments. Similarly, ensuring that records of intermediate and floating date data are kept in addition to the historically more common fixed date values enables greater flexibility to adapt data analysis procedures to



changing season length. Comprehensive documentation of glaciological field observations and consistent data and meta data

formats are essential for the preservation of historic time series, as well as for the quantification of future glacier change.





| Site | Data type | reference | doi |
|------|-----------|-----------|-----|
| MWK | Glacier wide ass balance and elevation zones, shapefile format | Stocker-Waldhuber et al. (2024a)* | 10.1594/PANGAEA.965660 |
| MWK | Glacier outlines, shapefile format | Stocker-Waldhuber et al. (2024c)* | 10.1594/PANGAEA.965626 |
| MWK | Point mass balance, tabular | Stocker-Waldhuber et al. (2024d)* | 10.1594/PANGAEA.965719 |
| MWK | Meteorological data, tabular | Stocker-Waldhuber et al. (2024b)* | 10.1594/PANGAEA.965646 |
| MWK | Glacier wide and elevation zone mass balance, tabular | Stocker-Waldhuber et al. (2016) | 10.1594/PANGAEA.806662 |
| VK | Glacier wide ass balance and elevation zones, shapefile format | Seiser et al. (2024a)* | 10.1594/PANGAEA.965648 |
| VK | Glacier outlines, shapefile format | Seiser et al. (2024c)* | 10.1594/PANGAEA.965619 |
| VK | Point mass balance, tabular | Seiser et al. (2024d)* | 10.1594/PANGAEA.965729 |
| VK | Meteorological data, tabular | Seiser et al. (2024b)* | 10.1594/PANGAEA.965647 |
| VK | Glacier wide and elevation zone mass balance, tabular | Seiser and Fischer (2016) | 10.1594/PANGAEA.833232 |

**Table 7.** Data publication series made available as part of this study. New data is added each season as it becomes available. The doi of the publication series leads to a landing page that lists all associated data sets. * denotes previously unpublished data made available as part of this study.

## 5    Data availability

Glaciological and meteorological data are uploaded to the pangaea repository (www.pangaea.de) each year and added to the publication series listed in Table 7

Glacier wide and elevation zone mass balance is also available through the WGMS database (WGMS, 2023) (https://doi.org/10.5904/wgms-520    fog-2023-09).

Code to process the glaciological and meteorological data and produce the figures in this publication is available at: https://github.com/LeaHartl/MWKVK_processing

*Author contributions.* All authors contributed to data collection and field work. BS and MS coordinate the mass balance measurements at VK and MWK, respectively. AF is the principal investigator for the mass balance programs at both sites. BS, MS, and LH processed the 525    mass balance data. AB, MVL, and LH processed the AWS data. LH produced the visualizations and statistics and wrote the paper with input from all coauthors.

*Competing interests.* We have no competing interests to declare.

*Acknowledgements.* The monitoring programs are funded by *Hohe Tauern National Park*, *Hydrographischer Dienst der Abteilung Wasserwirtschaft des Amtes der Tiroler Landesregierung* and *Hydrographischer Dienst des Landes Salzburg*. We thank the *Verein Gletscher Klima*



and numerous volunteer field work helpers who make the mass balance surveys possible. We are also very grateful for the support provided over the years by the teams of Kürsinger Hütte, Defregger Haus, and Johannis Hütte. Their support is invaluable. Matthias Huss (GLAMOS) provided generous advice on data structure and handling of uncertainties. Stefanie Schumacher and the PANGAEA team have supported our data publication process for many years. Thank you!



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

Glossary of glacier mass balance and related terms, 2011.

Collins, S. L.: Climate Change and Long-Term Ecological Research, BioScience, 72, 803–803, 2022.

Collins, S. L., Carpenter, S. R., Swinton, S. M., Orenstein, D. E., Childers, D. L., Gragson, T. L., Grimm, N. B., Grove, J. M., Harlan, S. L., Kaye, J. P., et al.: An integrated conceptual framework for long-term social–ecological research, Frontiers in Ecology and the Environment, 9, 351–357, 2011.

Costa, L., Hildén, M., Kropp, J., Böttcher, K., Fronzek, S., Swart, R., Otto, J., McCormick, N., Radojevic, M., Lückenkötter, J., et al.: Assessing climate impact indicators: Evaluation criteria and observed strengths and weaknesses, 2016.

Cremona, A., Huss, M., Landmann, J. M., Borner, J., and Farinotti, D.: European heat waves 2022: contribution to extreme glacier melt in Switzerland inferred from automated ablation readings, The Cryosphere, 17, 1895–1912, 2023.

Eder, K., Würländer, R., and Rentsch, H.: Digital photogrammetry for the new glacier inventory of Austria, International archives of pho-
togrammetry and remote sensing, 33, 254–261, 2000.

European Commission and Directorate-General for Environment and Sundseth, K.: The EU birds and habitats directives – For nature and people in Europe, Publications Office, https://doi.org/doi/10.2779/49288, 2015.

European Commission and Directorate-General for Environment and Sundseth, K.: The Habitats Directive – Celebrating 20 years of protecting biodiversity in Europe, Publications Office, https://doi.org/doi/10.2779/15019, 2012.

Fischer, A. and Kuhn, M.: Ground-penetrating radar measurements of 64 Austrian glaciers between 1995 and 2010, Annals of Glaciology, 54, 179–188, 2013.

Fischer, A., Stocker-Waldhuber, M., Seiser, B., Hynek, B., and Slupetzky, H.: Glaciological monitoring in Hohe Tauern National Park, eco.mont, 6, 2014.

Fischer, A., Seiser, B., Stocker-Waldhuber, M., and Abermann, J.: The Austrian Glacier Inventory GI 3, 2006, in ArcGIS (shapefile) format,
https://doi.org/10.1594/PANGAEA.844985, 2015a.

Fischer, A., Seiser, B., Waldhuber, M. S., Mitterer, C., and Abermann, J.: Tracing glacier changes in Austria from the Little Ice Age to the present using a lidar-based high-resolution glacier inventory in Austria, The Cryosphere, 9, 753–766, https://doi.org/10.5194/tc-9-753-2015, 2015b.

Fischer, A., Span, N., Kuhn, M., Helfricht, K., Stocker-Waldhuber, M., Seiser, B., Massimo, M., and Butschek, M.: Ground-penetrating radar
(GPR) point measurements of ice thickness in Austria, https://doi.org/10.1594/PANGAEA.849497, 2015.

Fischer, A., Patzelt, G., Achrainer, M., Groß, G., Lieb, G., Kellerer-Pirklbauer, A., and Bendler, G.: Gletscher im Wandel: 125 Jahre Gletschermessdienst des Alpenvereins., Springer Spektrum, 2018.

Gaiser, E. E., Bell, D. M., Castorani, M. C., Childers, D. L., Groffman, P. M., Jackson, C. R., Kominoski, J. S., Peters, D. P., Pickett, S. T., Ripplinger, J., et al.: Long-term ecological research and evolving frameworks of disturbance ecology, BioScience, 70, 141–156, 2020.



Geibel, L., Huss, M., Kurzböck, C., Hodel, E., Bauder, A., and Farinotti, D.: Rescue and homogenization of 140 years of glacier mass balance data in Switzerland, Earth System Science Data, 14, 3293–3312, 2022.

GLAMOS: Swiss Glacier Point Mass Balance Observations (release 2021), https://doi.org/10.18750/massbalance.point.2021.r2021, 2021.

Groß, G. and Patzelt, G.: The Austrian Glacier Inventory for the Little Ice Age Maximum (GI LIA) in ArcGIS (shapefile) format, PANGAEA, https://doi.org/10.1594/PANGAEA.844987, 2015.

Groß, G.: Der Flächenverlust der Gletscher in Österreich 1850-1920-1969, Zeitschrift für Gletscherkunde und Glazialgeologie, 23, 131–141, 1987.

Hansche, I., Fischer, A., Greilinger, M., Hartl, L., Hartmeyer, I., Helfricht, K., Hynek, B., Jank, N., Kainz, M., Kaufmann, V., Kellerer-Pirklbauer, A., Lieb, G., Mayer, C., Neureiter, A., Prinz, R., Reingruber, K., Reisenhofer, S., Riedl, C., Seiser, B., Stocker-Waldhuber, M., Strudl, M., Zagel, B., Zechmeister, T., and Schöner, W.: KryoMon. AT-Kryosphären Monitoring Österreich: 2021/22 Kryosphärenbericht
Nr. 1, https://doi.org/https://doi.org/10.25364/402.2023.1, 2023.

Hock, R. and Huss, M.: Glaciers and climate change, in: Climate Change, pp. 157–176, Elsevier, 2021.

Huemer, P.: Pseudo-endemism and cryptic diversity in Lepidoptera–case studies from the Alps and the Abruzzi, Journal on Protected Mountain Areas Research and Management, 3, 11–18, 2011.

Hugonnet, R., McNabb, R., Berthier, E., Menounos, B., Nuth, C., Girod, L., Farinotti, D., Huss, M., Dussaillant, I., Brun, F., et al.: Accelerated
global glacier mass loss in the early twenty-first century, Nature, 592, 726–731, 2021.

Huss, M. and Hock, R.: Global-scale hydrological response to future glacier mass loss, Nature Climate Change, 8, 135–140, 2018.

Huss, M., Bauder, A., and Funk, M.: Homogenization of long-term mass-balance time series, Annals of Glaciology, 50, 198–206, 2009.

Kaser, G., Fountain, A., Jansson, P., Heucke, E., and Knaus, M.: A manual for monitoring the mass balance of mountain glaciers, vol. 137, Unesco Paris, 2003.

Kuhn, M., Lambrecht, A., Abermann, J., Patzelt, G., and Groß, G.: The Austrian glaciers 1998 and 1969, area and volume changes, Zeitschrift für Gletscherkunde und Glazialgeologie, 43/44, 3–107, 2012.

Kuhn, M., Lambrecht, A., and Abermann, J.: Austrian glacier inventory 1998 (GI II), https://doi.org/10.1594/PANGAEA.809196, 2013.

Körner, C., Tappeiner, U., Newesely, C., Wittmann, H., Eberl, T., Kaiser, R., Meyer, E., Grube, M., Fernández Mendoza, F., Füreder, L., Niedrist, G. H., Daim, A., Lieb, G., Kellerer-Pirklbauer, A., Wickham, S., Petermann, J., and Berninger,
U.-G.: Langzeitmonitoring von Ökosystemprozessen im Nationalpark Hohe Tauern, Synthese der Startphase 2016-2018., https://doi.org/10.1553/GCP_LZM_NPHT_Synthese, 2020.

Lambrecht, A. and Kuhn, M.: Glacier changes in the Austrian Alps during the last three decades, derived from the new Austrian glacier inventory, Annals of Glaciology, 46, 177–184, https://doi.org/10.3189/172756407782871341, 2007.

Østrem, G. and Brugman, M.: Mass balance measurement techniques, A manual for field and office work, Environment Canada, Saskatoon,
600 1991.

Patzelt, G.: The Austrian glacier inventory: status and first results, IAHS Publication, 126, 181–183, 1980.

Patzelt, G.: Austrian glacier inventory 1969 (GI I), https://doi.org/10.1594/PANGAEA.807098, 2013.

Schaefli, B. and Huss, M.: Integrating point glacier mass balance observations into hydrologic model identification, Hydrology and Earth System Sciences, 15, 1227–1241, 2011.

Schütz, S. A. and Füreder, L.: Unexpected patterns of chironomid larval size in an extreme environment: a highly glaciated, alpine stream, Hydrobiologia, 820, 49–63, 2018.





Seiser, B. and Fischer, A.: Glacier mass balances and elevation zones of Venedigerkees, Hohe Tauern, Austria, 2011/2012 et seq., https://doi.org/10.1594/PANGAEA.833232, 2016.

Seiser, B., Stocker-Waldhuber, M., Hartl, L., Baldo, A., Lauria, M. V., and Fischer, A.: Glacier mass balance of Venedigerkees, 2011/12 et
seq., https://doi.org/10.1594/PANGAEA.965648, 2024a.

Seiser, B., Stocker-Waldhuber, M., Hartl, L., Baldo, A., Lauria, M. V., and Fischer, A.: Meteorological monitoring at LTER sites Venediger-
kees, 2019 et seq., https://doi.org/10.1594/PANGAEA.965647, 2024b.

Seiser, B., Stocker-Waldhuber, M., Hartl, L., Baldo, A., Lauria, M. V., and Fischer, A.: Glacier outlines of Venedigerkees, Austria, 2012 et
seq., https://doi.org/10.1594/PANGAEA.965619, 2024c.

Seiser, B., Stocker-Waldhuber, M., Hartl, L., Baldo, A., Lauria, M. V., and Fischer, A.: Point mass balance of Venedigerkees, Austria, 2011/12
et seq., https://doi.org/10.1594/PANGAEA.965729, 2024d.

Sommer, C., Malz, P., Seehaus, T. C., Lippl, S., Zemp, M., and Braun, M. H.: Rapid glacier retreat and downwasting throughout the European
Alps in the early 21st century, Nature communications, 11, 3209, 2020.

Stocker-Waldhuber, M., Helfricht, K., Hartl, A., and Fischer, A.: Glacier surface mass balance 2006–2014 on Mullwitzkees and Hallstätter
gletscher, Zeitschrift für Gletscherkunde und Glazialgeologie, 47, 101–119, 2013.

Stocker-Waldhuber, M., Fischer, A., and Kuhn, M.: Glacier mass balances and elevation zones of Mullwitzkees, Hohe Tauern, Austria,
2006/2007 et seq., https://doi.org/10.1594/PANGAEA.806662, 2016.

Stocker-Waldhuber, M., Seiser, B., Hartl, L., Baldo, A., Lauria, M. V., and Fischer, A.: Glacier mass balance of Mullwitzkees, 2006/07 et
seq., https://doi.org/10.1594/PANGAEA.965660, 2024a.

Stocker-Waldhuber, M., Seiser, B., Hartl, L., Baldo, A., Lauria, M. V., and Fischer, A.: Meteorological monitoring at LTER sites Mullwitz-
kees, 2020 et seq., https://doi.org/10.1594/PANGAEA.965646, 2024b.

Stocker-Waldhuber, M., Seiser, B., Hartl, L., Baldo, A., Lauria, M. V., and Fischer, A.: Glacier outlines of Mullwitzkees, Austria, 2012 et
seq., https://doi.org/10.1594/PANGAEA.965626, 2024c.

Stocker-Waldhuber, M., Seiser, B., Hartl, L., Baldo, A., Lauria, M. V., and Fischer, A.: Point mass balance of Mullwitzkees, Austria, 2006/07
et seq., https://doi.org/10.1594/PANGAEA.965719, 2024d.

Thibert, E., Blanc, R., Vincent, C., and Eckert, N.: Glaciological and volumetric mass-balance measurements: error analysis over 51 years
for Glacier de Sarennes, French Alps, Journal of Glaciology, 54, 522–532, 2008.

Vincent, C., Fischer, A., Mayer, C., Bauder, A., Galos, S. P., Funk, M., Thibert, E., Six, D., Braun, L., and Huss, M.: Com-
mon climatic signal from glaciers in the European Alps over the last 50 years, Geophysical Research Letters, 44, 1376–1383,
https://doi.org/10.1002/2016gl072094, 2017.

WGMS: Fluctuations of Glaciers Database, https://doi.org/10.5904/wgms-fog-2023-09, 2023.

Zemp, M., Thibert, E., Huss, M., Stumm, D., Rolstad Denby, C., Nuth, C., Nussbaumer, S. U., Moholdt, G., Mercer, A., Mayer, C., et al.:
Reanalysing glacier mass balance measurement series, The Cryosphere, 7, 1227–1245, 2013.

Zemp, M., Frey, H., Gärtner-Roer, I., Nussbaumer, S. U., Hoelzle, M., Paul, F., Haeberli, W., Denzinger, F., Ahlstrøm, A. P., Anderson, B.,
et al.: Historically unprecedented global glacier decline in the early 21st century, Journal of glaciology, 61, 745–762, 2015.

Zemp, M., Huss, M., Thibert, E., Eckert, N., McNabb, R., Huber, J., Barandun, M., Machguth, H., Nussbaumer, S. U., Gärtner-Roer, I., et al.:
Global glacier mass changes and their contributions to sea-level rise from 1961 to 2016, Nature, 568, 382–386, 2019.

Zemp, M., Eggleston, S., Míguez, B. M., Oakley, T., Rea, A., Robbez, M., and Tassone, C.: The status of the global climate observing system
2021: The GCOS status report, 2021.



Zemp, M., Gärtner-Roer, I., Nussbaumer, S. U., Welty, E. Z., Dussaillant, I., and Bannwart, J.: Global Glacier Change Bulletin No. 5 (2020-
2021), ISC (WDS) / IUGG (IACS) / UNEP / UNESCO / WMO, World Glacier Monitoring Service, Zurich, Switzerland, 2023.