# Peer review of "Fig. S1: Snow depth data at AWS\_MWK for the period of record. The red values represent an outlier filtered data set. No attempts were made to extract a meaningful signal after June 2022 due to the deterioration of the data quality."

_Earth System Science Data, 2023_

## Referee Comment (RC2)

**Review for Earth System Science Data**

Date: May/June 2014

Title: Glaciological and meteorological monitoring at LTER sites Mullwitzkees and Venedigerkees, Austria, 2006–2022

Authors: Lea Hartl, Bernd Seiser, Martin Stocker-Waldhuber, Anna Baldo, Marcela Violeta Lauria, and Andrea Fischer

In the submitted paper the authors present a very detailed and rich monitoring data set from glaciological mass balance and meteorological observation acquired during the last two decades at two glacier monitoring sites in the Austrian Alps. The data is accompanied by important insights into data acquisition, evaluation, and storage, as well as a transparent and well descried analysis of themselves. The manuscript is well structured, and the methodology follows an established approach (Geibel et al., 2022), which is further developed where necessary. Uncertainties are openly and transparently addressed, recognized and reflected in the data collection. The results are well described and convincing, accompanied by nice figures and lead to important findings for mass balance measurements and related data storage. The paper is already in a very good shape and almost ready to publish. I do not have major issues to correct, but some comments and questions (mainly on the figures) that may be addressed by the authors when preparing a next version.

L 68: Is it possible or even useful to show the outline of the Hohe Tauern National Park in Figure 1?

L 94: GI3 is the third glacier inventory of Austria? You could introduce this abbreviation here, as it will also be referred to later in the manuscript (e.g. in Fig. 1).

L 95: Do you know in which year Venedigerkees became disconnected from Obersulzbachkees?

L 123-124, **Figure 1**: The chapter 2.1 Study Sites and Figure 1 are closely linked together, when reading the chapter, it is of great help to consult Figure 1 in parallel. Figure 1 may be improved with (i) Showing the boundaries of Hohe Tauern National Park (if useful), (ii) adding a scale bar, (iii) indicating the height of Grossvenediger, (iv) indicating the height of the weather stations. The caption of the figure may be complemented with the full name of the glaciers: "Outlines of Venedigerkees (VK) and Mullwitzkees (MWK) for 2018 in red."

L 214-215, **Table 1**: I am not sure if the information in Table 1 is of great use like this. I can see that there is an enormous number of measurements available for both study sites and an average value per year is indicated. However, I have difficulties to bring the values from Table 1 in relation to Figure 2 and 3. Furthermore I don't know if the measurements are evenly distributed to the years (as the average value suggests), or if there is a bias.

L 214-215, **Figure 2** and **Figure 3**: This are important figures to show the dense monitoring networks at the two sites. However, it would be nice to see a little bit more about the context of the two glaciers in relation to their environment. It is maybe possible use contour lines or a hill shade of the relief to transport more information (e.g. to make obvious where steep or flat glacier parts are located). In addition, I think there is no reference to both figures in the text of the manuscript. Maybe you should add in the caption the statement "Coordinate reference system: MGI / Austria GK Central (EPSG:31255)" as you have it in Fig. 7ff.

L 245-249: You use the contour line method for spatial integration of the point mass balance. How is the expert knowledge gained that is incorporated? How many evaluators draw the lines? Did you ever compare results from different evaluators?

L 295: add "Table": …is provided in Table 4.

L 316-317; 331-332: Use italics when explaining the variables in the equation.

L 354-355: Should these values correspond with the values from Table 1?

L 372-373, **Figure 4**: I also like this Figure and showing the readings of 2022 in red is an enriching detail. For a better reading, I would like, if the panels could be labelled with a), b), and c). And I think it would be of value, to label the black vertical lines indicating April 30 and September 30 directly in the graph of panel a).

L 372-373, **Figure 5**: Maybe also here, labelling of the vertical line indicating September 30 would help to capture the timescale of the x-axis at first glance.
Maybe you could complement the titles of the panels with the height information of the stake? And would it be possible, to locate and name these three stakes in Figure2 and Figure 3 respectively, to know where these stakes are located?

L 374-275: This statement refers to counting's per year?

L 410: I guess the contour line at MWK are also manually derived? Maybe complement the sentence with this information.

L 421-422, **Figure 7** and **Figure 9**: Complement the caption of both figures with the information that you used contour lines to derive the zones of mass balance. What are the steps (in mm w.e.) between the different mass balance elevation zones?

L 421-422, **Figure 8** and **Figure 10**: The small points of snow depth probing are really hard to see. What are the steps (in mm w.e.) between the different mass balance elevation zones?

L 421-422, **Figure 7**, **Figure 8**, **Figure 9**, **Figure 10**: At first glance the label of the y-axis I interpreted the "m" as "E".

L 440-441, **Figure 12**: I really like this figure that gives an integrated and complete overview on the mass balance measurements and the related evaluation on both study sites. However, I think it's a pity to squeeze the results of both glaciers in into one figure. Wouldn't it be of additional value and easier to interpret the results, if you present it in two figures?

L 473-474, **Figure 13**, **Figure S3**: The readability of the figure would be improved by adding minor tick marks on the x-axis for the single months.

**Data Quality**: I was struggeling a bit, to download all the data related to the paper, as some were already easely accecible through PANGAEA and for others tokens and further links had to be used. However, at the end I think I managed to get all the data, but must say, that it is not easy to keep the overview on all the different packages. I hope this will be improved, as soon as the paper is accepted and the data published.

From all the ten data package provided via PANGAEA I checked at least one dataset. As far as I could see, the data sets are set up as explained in the paper (with quality flags and uncertainities) and seem to be complete and of great use for further studies.

References:

Geibel, L., Huss, M., Kurzböck, C., Hodel, E., Bauder, A., & Farinotti, D. (2022). Rescue and homogenisation of 140 years of glacier mass balance data in Switzerland. *Earth System Science Data Discussions*, *February* 1–30. https://doi.org/10.5194/essd-2022-56

---

## Author Comment (AC2)

Review for Earth System Science Data
Date: May/June 2014
Title: Glaciological and meteorological monitoring at LTER sites Mullwitzkees and Venedigerkees, Austria, 2006–2022
Authors: Lea Hartl, Bernd Seiser, Martin Stocker-Waldhuber, Anna Baldo, Marcela Violeta Lauria, and Andrea Fischer
In the submitted paper the authors present a very detailed and rich monitoring data set from glaciological mass balance and meteorological observation acquired during the last two decades at two glacier monitoring sites in the Austrian Alps. The data is accompanied by important insights into data acquisition, evaluation, and storage, as well as a transparent and well descried analysis of themselves. The manuscript is well structured, and the methodology follows an established approach (Geibel et al., 2022), which is further developed where necessary. Uncertainties are openly and transparently addressed, recognized and reflected in the data collection. The results are well described and convincing, accompanied by nice figures and lead to important findings for mass balance measurements and related data storage. The paper is already in a very good shape and almost ready to publish. I do not have major issues to correct, but some comments and questions (mainly on the figures) that may be addressed by the authors when preparing a next version.

Thank you for the encouraging and thoughtful review! We will be happy to address the comments. Please see below for responses to specific points.

L 68: Is it possible or even useful to show the outline of the Hohe Tauern National Park in Figure 1?
This is possible but perhaps not very useful. The national park includes an area considerably larger than the glaciers and showing the outline would require zooming out quite far, which makes it hard to see the locations of the weather stations, read annotations, etc. Due to this we would prefer not to include the park boundary. Here is a version of the figure showing the boundary in green for reference:

[Figure]

L 94: GI3 is the third glacier inventory of Austria? You could introduce this abbreviation here, as it will also be referred to later in the manuscript (e.g. in Fig. 1).
Yes, we will add an explanation of the abbreviation as suggested.

L 95: Do you know in which year Venedigerkees became disconnected from Obersulzbachkees?
We don't know with certainty. It is likely that the two tongues remained connected by a narrow band of debris-covered ice for some time after the 2012 boundaries were drawn. Whether this dead ice section constitutes a functional connection and when exactly this connection was severed cannot be determined exactly. We will add this information to the text.

L 123-124, Figure 1: The chapter 2.1 Study Sites and Figure 1 are closely linked together, when reading the chapter, it is of great help to consult Figure 1 in parallel. Figure 1 may be improved with (i) Showing the boundaries of Hohe Tauern National Park (if useful), (ii) adding a scale bar, (iii) indicating the height of Grossvenediger, (iv) indicating the height of the weather stations. The caption of the figure may be complemented with the full name of the glaciers: "Outlines of Venedigerkees (VK) and Mullwitzkees (MWK) for 2018 in red."
We will update the figure as suggested (minus the boundary of the national park, see above)

L 214-215, Table 1: I am not sure if the information in Table 1 is of great use like this. I can see that there is an enormous number of measurements available for both study sites and an average value per year is indicated. However, I have difficulties to bring the values from Table 1

in relation to Figure 2 and 3. Furthermore I don't know if the measurements are evenly distributed to the years (as the average value suggests), or if there is a bias.

We will adjust the caption of the table and associated text to explain this in more detail. Essentially, Fig. 2 and 3 show the network of measurement locations (the stakes move a little with the ice flow and sometimes have to be repositioned, hence they are not always in exactly the same place) and the table shows how often measurements were carried out at these locations. There is no bias in the sense of a temporal trend but the number of measurements and available stakes varies somewhat over the years. We can add a range of minimum and maximum numbers per year to the table.

L 214-215, Figure 2 and Figure 3: This are important figures to show the dense monitoring networks at the two sites. However, it would be nice to see a little bit more about the context of the two glaciers in relation to their environment. It is maybe possible use contour lines or a hill shade of the relief to transport more information (e.g. to make obvious where steep or flat glacier parts are located). In addition, I think there is no reference to both figures in the text of the manuscript. Maybe you should add in the caption the statement "Coordinate reference system: MGI / Austria GK Central (EPSG:31255)" as you have it in Fig. 7ff.

We will update the figures to include more topographic context (contours or hillshade), ensure they are mentioned in the text, and adjust the caption as suggested.

L 245-249: You use the contour line method for spatial integration of the point mass balance. How is the expert knowledge gained that is incorporated? How many evaluators draw the lines? Did you ever compare results from different evaluators?

Expert knowledge is gained by frequent visits to the sites by the same people over multiple years. This leads to an improved understanding of terrain dependent processes relevant to mass balance, e.g. knowledge of zones with recurring avalanche activity or wind deposition of snow during particular weather patterns. This in turn is helpful when interpreting ablation patterns, or in the case of anomalous stake readings (--> for example: could a specific outlier reading be due to an avalanche?) We believe that it is very beneficial if the person evaluating the mass balance and drawing the contours is familiar with the respective glacier and either knows the seasonal patterns over multiple years or can consult with people who have this experience.

In the first years of the monitoring program at MWK, up to four people did separate analyses with the contour line method. This resulted in differences of around ±100 mm w.e in the total, glacier wide mass balance. Due to the relatively minor differences the comparisons were later discontinued. The current approach is to aim for consistency by having the same person draw the contours every year (M. Stocker-Waldhuber for MWK, B. Seiser for VK). For an objective assessment of the contour line approach a comparison with geodetic mass balance is needed and will be carried out as soon as a suitable DEM becomes available.

L 295: add "Table": …is provided in Table 4.

Yes, we will add the missing word.

L 316-317; 331-332: Use italics when explaining the variables in the equation.
OK!

L 354-355: Should these values correspond with the values from Table 1?
Yes, they do correspond to the table in a general sense but they are grouped slightly differently in the text and table, e.g. the table gives the total number of intermediate measurements, while the text separates by measurement type (stakes, pits). We will rephrase this to clarify.

L 372-373, Figure 4: I also like this Figure and showing the readings of 2022 in red is an enriching detail. For a better reading, I would like, if the panels could be labelled with a), b), and c). And I think it would be of value, to label the black vertical lines indicating April 30 and September 30 directly in the graph of panel a).
We will add labels as suggested.

L 372-373, Figure 5: Maybe also here, labelling of the vertical line indicating September 30 would help to capture the timescale of the x-axis at first glance. Maybe you could complement the titles of the panels with the height information of the stake? And would it be possible, to locate and name these three stakes in Figure2 and Figure 3 respectively, to know where these stakes are located?
We will label the line and adjust Fig. 2 and 3 to indicate where the stakes are.

L 374-275: This statement refers to counting's per year?
Yes, we will rephrase to clarify.

L 410: I guess the contour line at MWK are also manually derived? Maybe complement the sentence with this information.
Yes, it is the same for both glaciers. We will add this to the sentence.

L 421-422, Figure 7 and Figure 9: Complement the caption of both figures with the information that you used contour lines to derive the zones of mass balance. What are the steps (in mm w.e.) between the different mass balance elevation zones?
We will adjust the captions and add the steps between the zones.

L 421-422, Figure 8 and Figure 10: The small points of snow depth probing are really hard to see. What are the steps (in mm w.e.) between the different mass balance elevation zones?
We will adjust the size of the markers for the probing and add the steps to the caption.

L 421-422, Figure 7, Figure 8, Figure 9, Figure 10: At first glance the label of the y-axis I interpreted the "m" as "E".
We will adjust the labels and spell out "meters"

L 440-441, Figure 12: I really like this figure that gives an integrated and complete overview on the mass balance measurements and the related evaluation on both study sites. However, I

think it's a pity to squeeze the results of both glaciers in into one figure. Wouldn't it be of additional value and easier to interpret the results, if you present it in two figures?
Thank you for the suggestion, we will turn this into two figures.

L 473-474, Figure 13, Figure S3: The readability of the figure would be improved by adding minor tick marks on the x-axis for the single months.
We will add tick marks!

Data Quality: I was struggeling a bit, to download all the data related to the paper, as some were already easely accecible through PANGAEA and for others tokens and further links had to be used. However, at the end I think I managed to get all the data, but must say, that it is not easy to keep the overview on all the different packages. I hope this will be improved, as soon as the paper is accepted and the data published. From all the ten data package provided via PANGAEA I checked at least one dataset. As far as I could see, the data sets are set up as explained in the paper (with quality flags and uncertainities) and seem to be complete and of great use for further studies.

All data will be publicly available (no tokens or additional links). We understand that the PANGAEA approach of having annual datasets grouped into publication series is sometimes not ideal if users simply want all the data. It does however allow for more flexible updates every year and provides DOI for single years/individual data sets, which can also have advantages. We will add some code examples to the github repo for the paper showing how to bulk download the data and generate the figures in the manuscript from the PANGAEA format.

---

## Author Response (AR1)

**Responses to Reviewer 1 (reviewer comments in black, responses in blue text):**
Having measured mass balance in the field for 40 years. It is unusual for me to come across a paper that provides significant useful approaches and insights. This paper looking at the mass balance of two glaciers in Austria does so. The authors have collected and analyzed a data set that is richer spatially and temporally during the balance year than other mass balance programs. The density of measurement through space and time limits extrapolation challenges and illustrates problems with a fixed date approach, where the fixed date is inflexible. The paper also highlights the important value of more frequent observations during the ablation season. I have a number of recommendations and questions below that could add value. I do not think these rise to the level of requiring an additional review.

Thank you sincerely for reviewing and for these encouraging comments! All of the measuring team was very happy to read this. The frequent measurements are a lot of work (as you know) and it is very motivating that others see value in these data.

Responses to specific comments are provided below.

25: reword "…drives hydrological change across spatial scales"

Changed as suggested.

44: Good point wonder if it is better worded with spatial and temporal resolution are included " Preserving all collected data with appropriate metadata captures the highest spatial and temporal resolution is essential for potential future reanalysis and homogenization of time series (Zemp et al., 2013)."

Changed as suggested.

78: Because you refer to a plateau and valley section, it is worth noting elevation range of these sections of MWK.

The flat and wide uppermost regions of the glacier above around 3150m are considered the plateau part of the glacier. This part is confined by a mountain ridge of the Hohe Zaun summit (3450m) and an ice divide at around 3200m to the northerly glacier Schlatenkees and to Frosnitzkees in the east. Below 3150m, the glacier steepens until it ends in a small, narrow and by now relatively short glacier tongue below 2900m. We added this information to the text.

81: the 50-70 m thickness in 2003 not particularly relevant by the end of the study period when ice has become much thinner. Any updated thickness values?

Unfortunately no updated thickness values. The emergence of rock outcrops in the upper sections indicates substantial losses. We hope to update the thickness data in the coming

years either with another GPR survey or geodetic MB as new surface elevation data become available.

92: On VK is the accumulation zone fed by avalanching or significant wind deposition?

Avalanches do not contribute significantly in the accumulation zone. Strong foehn winds from the south produce wind drift/deposition and likely some additional accumulation below the north face of Großvenediger.

144: Table 1 indicates an exceptionally high measurement density compared to the typical. This is worth pointing out as this also limits spatial extrapolation. The use of considerable fall probing is also something that is often not done reducing summer season measurement density.

We added a note in the 'outlook and conclusions' section to point out the advantages of high measurement density.

165-How consistent is snow density in snowpits near end of melt season? On many temperate glaciers late season snowpack has effectively a uniform density.

Density at the snow pits is mostly between around 370 and 500kg/m3, with seasonal variations and some elevation dependency. We aim to do the spring MB survey close to April 30 and have often found the snowpack at the lowest pit locations to be wet and mostly isothermal at this stage, while the snow is often still cold at higher locations.

[Figure]

238: The ELA in this case is an average location for the elevation where the accumulation zone begins, how patchy is the accumulation zone, which would indicate how useful the ELA as a separate measure from mass balance? On North Cascade glaciers I have found it impossible to report an observable/useful ELA and instead focus on reporting AAR to WGMS.

At VK, the glacier geometry is such that the ELA value is typically quite representative. There is some patchiness but overall the ELA indicates whether and how far ablation progresses into the upper basin below the north face of Großvenediger. At MWK, the elevation range with the greatest accumulation is shifted due to wind-drift from the plateau and, consequently, ablation can occur above the accumulation area. Here the ELA value is indeed not very useful and we will make a note on this in the revised manuscript. The AAR for both glaciers is reported to the WGMS along with the other data.

Below are some example images from the automatic camera showing the snow line at VK during end of summer conditions in multiple years. The accumulation zone of VK is on the left, the peak in the center of the image is Großvenediger.  2020-09-23:

[Figure]

2021-08-22:

[Figure]

2022-09-07:

[Figure]

This picture shows Mullwitzkees in 2016. The accumulation area is patchy and generally below the highest reaches of the glacier due to local wind effects.

[Figure]

360: Why is it considered essential to convert to a fixed date from the measurement observations?

This is required by the funding organizations of these particular monitoring programs, i.e. governmental hydrology agencies. They operate in the context of the hydrological year and request the glacier data to be reported per hydrological year.

Figure 4 Is an exceptional display of data. Particularly 4a. It would be relevant to use a specific year as an example as well.

Thank you! We adapted the figure to highlight the 2022 data in color.

Figure 5 illustrates ablation through time at specific stakes. Visually the rate of loss from year to year appears mostly consistent, statistically how consistent is it?

This depends on the stake and the time of year. Fig 6 shows annual ablation at the stakes (panels c and d) and the range of values recorded in different years. Generating representative statistics on subseasonal rates of loss is challenging since measurement intervals are irregular and the timing of snowmelt varies. Below are two plots attempting to visualize daily change rates for the intermediate measurements. This shows daily change rates for the same stakes as in Fig. 5 (i.e., the intermediate ablation values divided by the number of days between measurements).

[Figure]

[Figure]

This is a sine wave interpolation as per Zemp and Welty (2023) for a location where we have both a stake and a winter pit (same stake as panel a in the above plot):

394:  Not sure I see this as a challenge of this method.  I see this as a benefit of the method of such a high density of points. This limits spatial extrapolation which is the benefit of high density mass balance measurement programs.

Rephrased this so that both points (challenges inherent to probing and benefits of many points) are mentioned: "This represents only a relatively small fraction of the total probe points in the data sets, highlighting the challenges of the method and the importance of a high density of measurement points to capture spatial variability."

418:  Is this reduced ablation because of a higher albedo or simply excess accumulation due to the avalanches that remains after ablation conditions?

Both. Due to the avalanches, snow depth is locally higher and snow melt accordingly takes longer than at nearby stakes without avalanche input. Albedo is higher at the locations with

avalanche input for a longer amount of time since they are snow covered for a longer amount of time.

473: Overall did the AWS snow height sensor add any value?

Debatable... The VK AWS location is very wind exposed and the snow height at the AWS is not representative of general conditions. For the MWK AWS, we believe the snow height does represent "average" conditions in the area and problems were related to the sensor rather than the location. The MWK snow data may be helpful for future applications (the faulty sensor has been replaced) and statistical comparisons between the AWS data and the snow measurements on the glacier could be interesting once the AWS time series is a bit longer.

500: Why not note the advantages of moving to a system that tracks the balance year and is not fixed date?

Adjusted the text to more clearly state the advantages::

"The fixed date system has historically been the standard for glacier mass balance monitoring in Austria. However, as climatic conditions continue to change, temporal reanalysis of mass balance time series or adaptation of the measurement periods to a lengthened ablation season may become increasingly important to capture the actual seasonality of ablation and accumulation at the sites. Preserving intermediate mass balance measurements at the best available resolution along with key meteorological parameters is central to such efforts. While the fixed date system allows for comparison with historical data, the floating date system and frequent sub-seasonal measurements are advantageous in terms of reducing extrapolation uncertainties."

References:
Zemp, M., & Welty, E. (2023). Temporal downscaling of glaciological mass balance using seasonal observations. *Journal of Glaciology*, 1-6.

**Responses to Reviewer 2:**
In the submitted paper the authors present a very detailed and rich monitoring data set from glaciological mass balance and meteorological observation acquired during the last two decades at two glacier monitoring sites in the Austrian Alps. The data is accompanied by important insights into data acquisition, evaluation, and storage, as well as a transparent and well descried analysis of themselves. The manuscript is well structured, and the methodology follows an established approach (Geibel et al., 2022), which is further developed where necessary. Uncertainties are openly and transparently addressed, recognized and reflected in the data collection. The results are well described and convincing, accompanied by nice figures and lead to important findings for mass balance measurements and related data storage. The paper is already in a very good shape and almost ready to publish. I do not have major issues to correct, but some comments and questions (mainly on the figures) that may be addressed by the authors when preparing a next version.

Thank you for the encouraging and thoughtful review!
Please see below for responses to specific points.

L 68: Is it possible or even useful to show the outline of the Hohe Tauern National Park in Figure 1?
This is possible but perhaps not very useful. The national park includes an area considerably larger than the glaciers and showing the outline would require zooming out quite far, which makes it hard to see the locations of the weather stations, read annotations, etc. Due to this we would prefer not to include the park boundary. Here is a version of the figure showing the boundary in green for reference:

[Figure]

L 94: GI3 is the third glacier inventory of Austria? You could introduce this abbreviation here, as it will also be referred to later in the manuscript (e.g. in Fig. 1).
We added an explanation of the abbreviation as suggested.

L 95: Do you know in which year Venedigerkees became disconnected from Obersulzbachkees?
We don't know with certainty. It is likely that the two tongues remained connected by a narrow band of debris-covered ice for some time after the 2012 boundaries were drawn. Whether this dead ice section constitutes a functional connection and when exactly this connection was severed cannot be determined exactly. We added a short note on this in the text.

L 123-124, Figure 1: The chapter 2.1 Study Sites and Figure 1 are closely linked together, when reading the chapter, it is of great help to consult Figure 1 in parallel. Figure 1 may be improved with (i) Showing the boundaries of Hohe Tauern National Park (if useful), (ii) adding a scale bar, (iii) indicating the height of Grossvenediger, (iv) indicating the height of the weather stations. The caption of the figure may be complemented with the full name of the glaciers: "Outlines of Venedigerkees (VK) and Mullwitzkees (MWK) for 2018 in red."
We updated the figure as suggested (minus the boundary of the national park, see above)

L 214-215, Table 1: I am not sure if the information in Table 1 is of great use like this. I can see that there is an enormous number of measurements available for both study sites and an average value per year is indicated. However, I have difficulties to bring the values from Table 1 in relation to Figure 2 and 3. Furthermore I don't know if the measurements are evenly distributed to the years (as the average value suggests), or if there is a bias.
Essentially, Fig. 2 and 3 show the network of measurement locations (the stakes move a little with the ice flow and sometimes have to be repositioned, hence they are not always in exactly the same place) and the table shows how often measurements were carried out. There is no bias in the sense of a temporal trend but the number of measurements and available stakes varies over the years. We have adjusted the text to clarify that stakes do not remain in exactly the same place and added the range of minimum and maximum number of measurements per year to the table.

L 214-215, Figure 2 and Figure 3: This are important figures to show the dense monitoring networks at the two sites. However, it would be nice to see a little bit more about the context of the two glaciers in relation to their environment. It is maybe possible use contour lines or a hill shade of the relief to transport more information (e.g. to make obvious where steep or flat glacier parts are located). In addition, I think there is no reference to both figures in the text of the manuscript. Maybe you should add in the caption the statement "Coordinate reference system: MGI / Austria GK Central (EPSG:31255)" as you have it in Fig. 7ff.

We updated the figures to include a hillshade layer and adjusted the caption as suggested.

L 245-249: You use the contour line method for spatial integration of the point mass balance. How is the expert knowledge gained that is incorporated? How many evaluators draw the lines? Did you ever compare results from different evaluators?
Expert knowledge is gained by frequent visits to the sites by the same people over multiple years. This leads to an improved understanding of terrain dependent processes relevant to mass balance, e.g. knowledge of zones with recurring avalanche activity or wind deposition of snow during particular weather patterns. This in turn is helpful when interpreting ablation patterns, or in the case of anomalous stake readings (--> for example: could a specific outlier reading be due to an avalanche?) We believe that it is very beneficial if the person evaluating the mass balance and drawing the contours is familiar with the respective glacier and either knows the seasonal patterns over multiple years or can consult with people who have this experience.

In the first years of the monitoring program at MWK, up to four people did separate analyses with the contour line method. This resulted in differences of around ±100 mm w.e in the total, glacier wide mass balance. Due to the relatively minor differences the comparisons were later discontinued. The current approach is to aim for consistency by having the same person draw the contours every year (M. Stocker-Waldhuber for MWK, B. Seiser for VK). For an objective assessment of the contour line approach a comparison with geodetic mass balance is needed and will be carried out as soon as a suitable DEM becomes available.

L 295: add "Table": …is provided in Table 4.
Yes, added the missing word.

L 316-317; 331-332: Use italics when explaining the variables in the equation.
Adjusted to use italics.

L 354-355: Should these values correspond with the values from Table 1?
Yes, they correspond to the table but are grouped slightly differently in the text and table, i.e., the table gives the total number of intermediate measurements, while the text separates by measurement type (stakes, pits).

L 372-373, Figure 4: I also like this Figure and showing the readings of 2022 in red is an enriching detail. For a better reading, I would like, if the panels could be labelled with a), b), and c). And I think it would be of value, to label the black vertical lines indicating April 30 and September 30 directly in the graph of panel a).
Highlighted 2022 readings in red and added labels as suggested.

L 372-373, Figure 5: Maybe also here, labelling of the vertical line indicating September 30 would help to capture the timescale of the x-axis at first glance. Maybe you could complement the titles of the panels with the height information of the stake? And would it be possible, to locate and name these three stakes in Figure2 and Figure 3 respectively, to know where these stakes are located?
Added labels as suggested. Adjusted Fig. 2 and 3 to indicate where the stakes are.

L 374-275: This statement refers to counting's per year?
Yes, rephrased to clarify.

L 410: I guess the contour line at MWK are also manually derived? Maybe complement the sentence with this information.
Yes, it is the same for both glaciers, added a note to clarify this in the text.

L 421-422, Figure 7 and Figure 9: Complement the caption of both figures with the information that you used contour lines to derive the zones of mass balance. What are the steps (in mm w.e.) between the different mass balance elevation zones?
Adjusted the captions to explain the steps between the zones. (250 to 500 mm w.e. steps for annual MB).

L 421-422, Figure 8 and Figure 10: The small points of snow depth probing are really hard to see. What are the steps (in mm w.e.) between the different mass balance elevation zones?
Increased the marker size for the probing points and added a note on the steps to the captions (200 mm w.e. steps for winter MB).

L 421-422, Figure 7, Figure 8, Figure 9, Figure 10: At first glance the label of the y-axis I interpreted the "m" as "E".
Adjusted axis labels to spell out "meters"

L 440-441, Figure 12: I really like this figure that gives an integrated and complete overview on the mass balance measurements and the related evaluation on both study sites. However, I think it's a pity to squeeze the results of both glaciers in into one figure. Wouldn't it be of additional value and easier to interpret the results, if you present it in two figures?
Thank you for the suggestion, we turned this into two figures.

L 473-474, Figure 13, Figure S3: The readability of the figure would be improved by adding minor tick marks on the x-axis for the single months.
Added monthly tick marks as suggested.

Data Quality: I was struggling a bit, to download all the data related to the paper, as some were already easely accecible through PANGAEA and for others tokens and further links had to be used. However, at the end I think I managed to get all the data, but must say, that it is not easy to keep the overview on all the different packages. I hope this will be improved, as soon as the paper is accepted and the data published. From all the ten data package provided via PANGAEA I checked at least one dataset. As far as I could see, the data sets are set up as explained in the paper (with quality flags and uncertainities) and seem to be complete and of great use for further studies.

All data will be publicly available (no tokens or additional links). We understand that the PANGAEA approach of having annual datasets grouped into publication series is sometimes not ideal if users simply want all the data. It does however allow for more flexible updates every year and provides DOI for single years/individual data sets, which can also have advantages. We have added code examples to the github repo for the paper explaining how to bulk download the data and generate figures in the manuscript from the PANGAEA format.

---

## Author Response (AR2)

Dear Dr. Hartl and co-authors,

Thank you for this revised version and for addressing the reviewers' comments.

The reviewers confirmed that your dataset is valuable and well-structured and that your article was well written despite the amount and complexity of the data that you have to present.

I still have some comments on your tables and figures, some of which I still find hard to read. Once you have addressed those comments, I will be happy to accept your manuscript for publication without further review.

Kind regards,
Baptiste Vandecrux

Dear Dr. Vandecrux, thank you for the thoughtful comments and time spent on this. I think we have addressed all the points, see details below.
Lea Hartl on behalf of the authors

Figure 1: Please harmonize the way you present elevation: make all text labels black, show names of AWS (AWS_VK and AWS_MWK) next to their markers and give elevation under the name without parentheses. For G, you could consider adding a black cross at the summit location.
Changed as suggested.

Table 2: I am wondering if the table would be clearer if transposed. It also starts to be many values in individual cells. Consider adding more columns (in its current form) or rows, if transposed (sorry for the text formatting):
...
Transposed the table. This does make it easier to read, good idea.

Figure 2: Please move the legend so that it does not hide the axis. The legend could be made more complete and clearer by grouping markers under categories:

GI 1-3 should be replaced by their years and shown with the same solid lines as outlines for 2012-2022. All outlines should be grouped under a "Glacier outlines" category. You can mention in the caption that the data source for each outline is given in Table 2. Why not plot all outlines from Table 2 in the Figure? If two outlines are overlapping, you can mention it in the caption. In the legend again, colored circles should be placed under a "Point measurement" category, which should also present (before the years) a black circle for "stake" and a black square for "snow pit". All the yearly markers in the legend could be replaced by a small vertical color bar with label "2007" next to the top and "2022" next to the

bottom. "AWS" in the legend could be replaced by "AWS_MWK" so it doesn't have to be explained in the caption.

Figure 3: Same as Figure 2. For this Figure, "stake 95" and "stake 100/VEK-24" could be given as labels on the map next to the black circle. Maybe a single circle/oval could be used to indicate each of these stakes' areas (since they change location with the year). The same could be done in Figure 2 for stake 11.

Adjusted the figures to address these points. Fig 2 and 3 now include all outlines listed in Table 2. The initial intent was to focus the figures on the newly published outlines, but showing all of them is of course also possible. GI4 (2015) is dashed because this one has some inconsistencies with the others due to a different mapping approach and to distinguish it from the other 2015 outline available for MWK. We note this in the caption. Zooming out far enough to show the full extent of the historic outlines makes it hard to distinguish the stake and pit locations so I left the zoom level as it was.

Table 4: Make range and accuracy separate columns. Make "Wind speed" and "wind direction" two separate rows that share the same instrument. Remove "Wind Speed & Direction Sensor" from the sensor name cell, as it is completely redundant with the parameter name. Make "Shortwave radiative flux" and "Longwave radiative flux" separate rows that share the same sensor. The calibration date and derived uncertainty do not fit with the Range and Accuracy columns of the other parameters. Please move that information to the main text and explain what the reported uncertainty means or how it should be used. The table caption should describe what is in the table, but "Data logger at both stations: Campbell Scientific CR3000." is information of its own. Please move it to the text.

Made separate rows as suggested and moved information re. calibration and data logger to the text. I am struggling with the formatting of this table in the latex template and did not manage to nicely format the rows with multi-row cells. The cells are all in the latex table as they should be so it is a matter of centering the text and placing lines. I can't figure out how to do that for the multi rows. I am guessing that the table formatting will be adjusted to match the journal layout requirements and I would be very grateful if this could be handled at that stage.

Table 5: You tend to use lines to report multiple pieces of information: [parameter + site] or [% flagged for SWin + SWout]. This makes the table more complicated to read. Please make separate rows for wind speed and wind direction, for downwelling/upwelling shortwave/longwave radiation. Please move "low temperature flags" to the caption or to a note at the bottom of the table. I strongly suggest that you make two columns "% flagged for AWS_MWK" and "% flagged for AWS_VK" and have one row per parameter. The Start/End dates were very hard to read and understand due to the grouping per parameter instead of per station. It would be much clearer to say that "AWS_MWK was active between

2020-08-10 and 2023-10-31 (except for radiation which started on 2021-09-21 and snow height which is available between 2019-07-01 and 2022-06-02)." and "AWS_VK was active between 2019-09-19 and 2023-10-31 (except for snow height which is available between 2020-07-01 and 2022-06-28)." This information should appear clearly in the text (instead of a very unspecific "... for a sub-period of the overall time series" on l.333), and if you think it is important, in the caption of Table 5. I feel that if the dates are clear in the text, they do not need to appear in Table 5.

Changed the table as suggested, moved the date information to the text and rephrased to make this more clear. The stations are still active, hence we indicate that the percentages given here are for the data up to October 1, 2023 (i.e., the same as the data set currently available on pangaea)

Figure 4 and 5: Please make sure that "July 1" and "September 30" do not overlap with the figures' frames.

Moved annotations further away from the figure frame.

Figure 7 & 8: A lot of plotting space is lost between the panels. I strongly suggest you update these figures so that the year is inside each panel on the bottom left or center and that the axis ticks point inward. Then the space between subplots can be removed altogether. Optional: make a legend showing the different types of markers and place it to the right of the 2022 panel.

Changed as suggested.

Figure 8: Please increase the size of the probing circles (unless my previous comment makes them visible enough). Give an edge color to them.

Increased size and added a red edge.

Figure 9 & 10: Same as Figure 7 & 8. Please place the year in the top-right corner of each panel and remove space between subplots. A single-line legend could be placed at the top.

Changed as suggested.

Figure 12 & 13: Optional: You could put the year in the top left corner and remove the spaces between subplots. This would allow putting both figures on the same page.

Moved the year to the corner and reduced white space. The latex template still does not place the figures on one page but this would probably be possible in layouting. I did not want to mess with the settings in the template so I did not attempt changing the fig size.

Figure 14 and S3: I cannot distinguish daily mean RH and daily mean pressure. Please plot the two variables in separate panels. For temperature, I do not see what "daily low/high, monthly mean" can be used for. Is it used in the text? If not, it can be removed from the graph. For all panels, make the legend a single line (except for radiation where there is

space) and modify the y-axis to ensure that the legend does not mask the data. Please plot longwave radiation as well.

Added panels to separate RH and pressure and to show longwave radiation. Removed lines for mean monthly daily low/high temperatures (this can be of interest in a climatological context but is not needed here). Changed legend positioning as suggested.

I strongly suggest bringing Figure S3 into the manuscript as Figure 15 and removing Table 6, which doesn't give more information than Figure 14 and S3. The first paragraph of Section 3.2 can be supported by Figure 14 and S3 without problem.

Fig. S3 is now Fig 15 in the manuscript. I moved Table 6 into the supplement as Table S1.

Table 7: Please make the DOIs clickable hyperlinks. Make sure "DOI" is capitalized in the entire article. Capitalize "Reference" just like the other column headers.

Changed as suggested.

Figure S1 and S2: Great work to remove all this noise. I am wondering if the plotted data are hourly averages of 10-minute values. If that is the case, I am wondering whether you are de-noising the 10-minute data or directly the hourly averages. I have seen that type of noise in Greenland AWS data. It appeared when the sonic rangers "see" two reflectors (one being the surface the other one that could be the station mast or a cable). The hourly average then jumps based on how many times the wrong reflector was picked during that hour. Going back to the highest temporal resolution should show those two separate reflectors without the mixing induced by temporal averaging and should make the removal of the spurious reflector easier. This does not have to be implemented for the publication of this article.

The black markers in the plots are 10 minute values as recorded by the logger and this is what we are using for the denoising, which involves a lot of pretty heavy handed smoothing. In our case the multiple "lines" of spurious data are not a result of hourly averaging, this is already present in what we have from the logger. We think it is related to an issue with the actual sensor rather than a signal from a "wrong" reflector but have not been able to pin point the problem with certainty. The logger samples every minute and stores 10 minute averages, so it is possible that the mixing issue you describe happens at that stage - we will see if we can change the logger settings to check if that is causing it (this is beyond the scope of this article but interesting for future work with these stations).

Figure 15 and Lines 451-454: Present the PDD sums for the two stations. Since PDD is not a variable present in your dataset, it falls outside of the scope of an ESSD article, which should only describe the dataset. Please either remove or boil it down to a single sentence stating which years had the highest/lowest numbers of positive degree days. Figure 15 should be removed anyway.

Removed the figure and the text related to PDD.

l. 514: "are advantageous in terms of" replace with "have the advantage of"

Replaced as suggested